# Cdc14 phosphatase counteracts Cdk-dependent Dna2 phosphorylation to inhibit resection during recombinational DNA repair

Adrián Campos [1,3], Facundo Ramos [1,3], Lydia Iglesias [1], Celia Delgado [1], Eva Merino[1], Antonio Esperilla-Muñoz[2], Jaime Correa-Bordes [2] & Andrés Clemente-Blanco [1] ✉

Cyclin-dependent kinase (Cdk) stimulates resection of DNA double-strand breaks ends to generate single-stranded DNA (ssDNA) needed for recombinational DNA repair. Here we show in *Saccharomyces cerevisiae* that lack of the Cdk-counteracting phosphatase Cdc14 produces abnormally extended resected tracts at the DNA break ends, involving the phosphatase in the inhibition of resection. Over-resection in the absence of Cdc14 activity is bypassed when the exonuclease Dna2 is inactivated or when its Cdk consensus sites are mutated, indicating that the phosphatase restrains resection by acting through this nuclease. Accordingly, mitotically activated Cdc14 promotes Dna2 dephosphorylation to exclude it from the DNA lesion. Cdc14-dependent resection inhibition is essential to sustain DNA re-synthesis, thus ensuring the appropriate length, frequency, and distribution of the gene conversion tracts. These results establish a role for Cdc14 in controlling the extent of resection through Dna2 regulation and demonstrate that the accumulation of excessively long ssDNA affects the accurate repair of the broken DNA by homologous recombination.

DNA repair of double-strand breaks (DSBs) is central to the maintenance of genome integrity in all eukaryotic organisms. To maintain genomic stability in response to a DSB, cells have evolved conserved and sophisticated DNA repair mechanisms that guarantee the reconstitution of the DNA molecule. One of the most reliable routes to repair a DSB is the so-called homologous recombination (HR) pathway, a complex mechanism that relies on the search for a similar or identical intact DNA sequence to copy the lost genetic information. HR always entails the processing of the DSB by removing nucleotides of the 5′-end of the strand to generate a long 3′ single-stranded DNA (ssDNA) in a process known as resection[1]. Resection is performed in a two-step sequential manner. In the first step, the nuclease activity of Mre11 within the MRX complex (Mre11-Rad50-Xrs2) together with Sae2, nicks the strand to be resected and degrades nucleotides in the 3′→5′ direction towards the DSB, generating a short 3′ overhang ssDNA[2,3]. In the second step, the exonuclease Exo1 and the Sgs1-Dna2-Top3-Rmi1 complex recognize this structure and trigger 5′→3′ long-range resection that ends up with the generation of extensive ssDNA tracts[4,5]. The formation of long ssDNAs at both sides of the DNA lesion is essential to search for and invade a homologous DNA sequence that serves as donor for repair, generating a structure named the displacement loop (D-loop). Once the lost information has been copied, the D-loop is

[1]Cell Cycle and Genome Stability Group, Instituto de Biología Funcional y Genómica (IBFG), CSIC-USAL, Salamanca, Spain. [2]Departamento de Ciencias Biomédicas, Universidad de Extremadura, Badajoz, Spain. [3]These authors contributed equally: Adrián Campos, Facundo Ramos. ✉e-mail: andresclemente@usal.es

disengaged and the newly copied DNA is re-annealed to the broken chromosome, where the 3′ ends prime DNA polymerization to fill the gaps in both directions from the DSB in a process known as DNA re-synthesis. The execution of this recombinational DNA repair pathway always involves the unidirectional transfer of DNA from the donor to the recipient sequence, resulting in the formation of gene conversion (GC) events.

Several kinases have been implicated in the correct orchestration of the DNA damage response (DDR), including Mec1, Rad53, and Chk1; the mammalian homologs of ATR, CHK2 and CHK1 respectively. In recent years, it has become clearer that the cyclin-dependent kinase (Cdk) is also essential for the correct execution of several stages of the DNA repair pathway[6], mainly by stimulating early DNA repair events. In this regard, it has been demonstrated that Cdk-dependent phosphorylation of Sae2 and Mre11 enhances short-range resection, biasing DNA repair to HR[7–9]. Besides, Cdk phosphorylation of Dna2 triggers long-range resection by stimulating its transport from the cytoplasm to the nucleus[10], as well as its binding to the DSB[11]. Although the activation of resection is probably one of the most important and well-described functions attributed to Cdk, this kinase has also been implicated in later repair events, as in the recruitment of the recombinational protein Rad52 to damage sites[12] and in the correct activation of the Srs2 helicase during recombinational DNA repair[13].

Cdc14 is a highly conserved phosphatase within the DUSPs (Dual-specific phosphatases) family that is characterized by its ability to dephosphorylate both phosphotyrosine and phosphoserine/phosphothreonine residues in its substrates. Cdc14 was originally described for its ability to dephosphorylate proteins previously phosphorylated by Cdk during mitotic exit in *Saccharomyces cerevisiae*[14]. Further studies on Cdc14 orthologues from yeast to humans have identified additional roles for this family of phosphatases in multiple cellular processes, including cytokinesis[15–17], transcription[18–20], centrosome duplication[21–23] and chromosome segregation[24–29]. Although most of these functions seem to be conserved along evolution in different species, some of the roles attributed to Cdc14 are not shared by different organisms[30].

In *S. cerevisiae*, the activity of Cdc14 is regulated by the cell cycle-dependent association with its nucleolar competitive inhibitor Net1[31]. During interphase, the phosphatase is sequestered in the nucleolus, thus allowing Cdk to phosphorylate multiple nuclear targets. When cells reach mitosis, two distinctive pathways named FEAR (Fourteen Early Anaphase Release) and MEN (Mitotic Exit Network) release the protein from the nucleolus into the nucleoplasm and cytoplasm in early and late anaphase, respectively, triggering dephosphorylation of multiple Cdk substrates[32]. Interestingly, Cdc14 is accumulated in the nucleoplasm in the presence of genotoxic drugs during DDR activation in the budding and fission yeasts[33–35], suggesting a role for the phosphatase in the response to DNA damage. Supporting this observation, in recent years it has become evident that DDR-induced Cdc14 participates in the correct orchestration of different stages encompassed in the DNA damage response in several model organisms[34–38]. Considering the vast number of Cdk targets identified in response to DNA damage, it is tempting to speculate that Cdc14 might regulate the DDR by reversing Cdk-dependent phosphorylation of proteins involved in the repair of a DNA lesion[39].

Here we show in *S. cerevisiae* that Cdc14 activity is required for neither resection proficiency nor for checkpoint activation in response to an HO endonuclease-induced DSB. However, cells lacking Cdc14 develop a marked over-resection of DSB ends, implicating the phosphatase in the counteraction of resection activity. Over-resection of DSBs in the absence of the phosphatase is bypassed when removing Dna2 or when expressing a Cdk phospho-deficient version of the nuclease, demonstrating that Cdc14 controls resection inhibition by targeting Dna2. Accordingly, mitotically activated Cdc14 promotes Dna2 dephosphorylation to impede its binding to

DSBs, thus restraining resection progression. Genome-wide sequencing of cells lacking Cdc14 activity revealed defects in DNA re-synthesis at the DNA break that disturb the restoration of the DNA molecule. Finally, we report a global genomic approach based on the analysis of discordant read pairs to demonstrate that Cdc14 is required for the correct distribution of GC products during recombinational DNA repair.

Overall, we propose that the tight coordination between Cdk and Cdc14 activities during the damage response is required to ensure ideal extension of the ssDNA tracts generated during resection to facilitate the execution of subsequent stages of the HR pathway.

## Results

### Cdc14 is required for inter-chromosomal DNA repair by homologous recombination

To fully characterize the role of Cdc14 phosphatase in the repair of DNA lesions, we decided to use the PMV background of *Saccharomyces cerevisiae*, a system that allows the use of genome-wide sequencing to measure multiple parameters of the repair process accurately[40]. In this strain, the endogenous HO promoter has been substituted for the inducible *GAL1* promoter, so the addition of galactose to the culture generates a DSB at the *MATa locus* at chromosome III. The *HMR* and *HML* sequences have been deleted to avoid intra-chromosomal DNA repair after HO cleavage (Fig. 1a). As bait for HR, the PMV contains a *MATa′* sequence inserted into the *ARG5,6* locus of chromosome V. This sequence harbors 23 polymorphisms each separated by approximately 60 nucleotides (nt) across the two 0.7 kb regions flanking an uncleavable HO-*inc* site (Fig. 1a). The presence of these polymorphisms allows the analysis of DSB repair by HR in a genome-wide sequencing approach[40]. Three of the polymorphisms included in the *MATa′* donor sequence were designed to generate *Eco*RI restriction sites at both sides of the break, thus facilitating the analysis of the length and directionality of GC events taking place during the repair of the HO-induced break by physical Southern blot (Fig. 1b).

To dissect the possible roles of Cdc14 in recombinational DNA repair, we substituted the endogenous *CDC14* allele for the thermo-sensitive variant *cdc14-1* in the PMV strain and checked for DNA repair deficiencies in Southern blot experiments. To rule out possible cell cycle interferences when working with cells lacking Cdc14 activity, we pre-synchronized both wild-type and *cdc14-1* strains in G1 by using alpha factor before releasing them into fresh media without the pheromone. One hour after the release, we transferred cultures to 33 °C to inactivate the phosphatase, added galactose to the media to induce the HO expression, and then took samples every hour for analysis. FACS profiles showed that both strains were arrested with similar kinetics with 2c content due to checkpoint activation (Supp. Fig. 1a). Accordingly, Rad53 was phosphorylated in a timely manner in both backgrounds (Supp. Fig. 1b), indicating that Cdc14 is not required for activation of the DNA damage response in the PMV strain. Six hours after HO induction, wild-type cells initiated cell cycle reentry after repair, as denoted by the increase in 1c cells observed in the FACS profile (Supp. Fig. 1a), the appearance of dephosphorylated forms of Rad53 (Supp. Fig. 1b), and the accumulation of anaphase/G1 cells (Supp. Fig. 1c). However, the *cdc14-1* mutant remained arrested in 2c for the entire experiment (Supp. Fig. 1a), with high levels of Rad53 phosphorylation (Supp. Fig. 1b) and with most cells blocked in G2/M (Supp. Fig. 1c), indicating a permanent DNA damage checkpoint arrest.

To analyze the dynamics of HO repair in wild-type and *cdc14-1* mutant, we performed Southern blot analysis using two probes that hybridize at both sides of the *MATa locus* (Fig. 1a). Hybridization with the left probe showed a drastic decrease in the uncut band signal (6.8 kb) and an accumulation of the cut product (2.5 kb) by 1 h after HO induction, indicating that both strains generated the

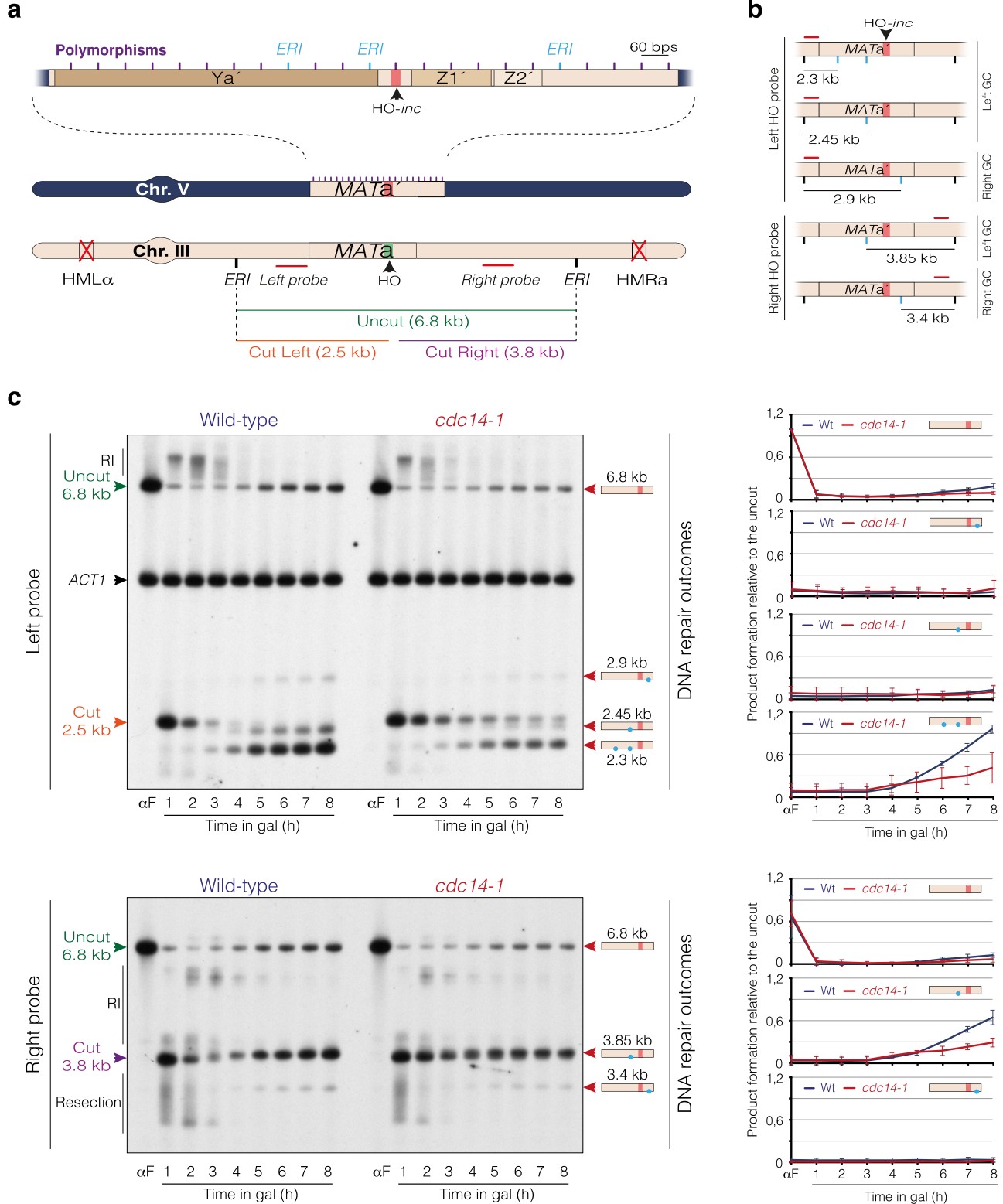

HO-induced break with the same efficiency (Fig. 1c top panel). We also detected a similar initial resection rate in both strains as denoted by the similar dynamics in the disappearance of the cut product and a comparable profile in the accumulation of resection intermediates associated with the uncut signal (Fig. 1c top panel, RI). However, lack of Cdc14 resulted in a drastic reduction in the accumulation of all repair products generated during the repair of the HO-induced break. Still, the relative proportion of all repair outcomes was similar between the strains (Fig. 1c top panel, DNA

repair outcomes), indicating that as in the wild-type, GC in the *cdc14-1* mutant preferentially occurred to the left side of the HO-induced break. All these observations were recapitulated when using a probe right to the HO cleavage site, including the efficiency in HO cutting (Fig. 1c bottom panel, 3.8 kb band), the resection rate (Fig. 1c bottom panel, 3.8 kb band and RI), and GC (Fig. 1c bottom panel, DNA repair outcomes). These results indicate that Cdc14 is not required for triggering DNA end resection, but it is required for later stages of the recombinational DNA repair pathway.

**Fig. 1 | Cdc14 phosphatase is required for recombinational DNA repair.**
**a** Schematic representation showing the relevant genomic structure of the PMV strain. The *HML*α and *HMR*a *loci* at chromosome III have been deleted to drive *MAT*a recombination to a 1.3 kb *MAT*a' fragment located at the *ARG5,6 locus* in chromosome V that contains 23 polymorphisms (purple lines). Three of the *MAT*a' polymorphisms generate *Eco*RI sites (*ERI*) at both sides of the uncleavable HO-*inc* site (blue lines), facilitating the analysis of GC by Southern blot experiments. The size of the "Uncut" and "Cut" DNA fragments generated after inducing the HO endonuclease are shown. The probes used to determine the incorporation of the *Eco*RI sites from chromosome V into III by Southern blot are shown. **b** Diagram representing the hypothetical GC outcomes that could be generated during the repair of the HO-induced break. Horizontal red lines represent the position of the left and right probes. Vertical black lines represent the *Eco*RI sites on chromosome III. Vertical blue lines denote the potential *Eco*RI sites incorporated from the *MAT*a' into the *MAT*a *locus* after GC. The size of the bands generated when digesting with *Eco*RI for each possible GC outcome is shown. **c** Southern blot analysis of DNA repair in wild-type and *cdc14-1* PMV cells. The strains were grown in YP-Raffinose at 25 °C and alpha factor (αF) was added to block the cells in G1. After the arrest, cells were liberated and galactose was added to the media to induce HO expression. One hour after HO induction, cultures were transferred to 33 °C and samples were taken at the indicated time points. DNA was extracted, digested with *Eco*RI and analyzed by Southern blot using the left (top panel) and right (bottom panel) probes shown in **a**. The DNA molecular weights on the left correspond to the distinctive DNA fragments shown in **a**. An *ACT1* gene sequence was used as a loading control. RI: resection intermediates. Diagrams on the right represent *Eco*RI sites (blue dots) surrounding the HO cleavage site (red bar) transferred from the *MAT*a' to the *MAT*a *locus* as depicted in **b**. The graphs on the right represent the quantification of the different GC outcomes obtained after repair. The mean ± SD from three independent Southern blots experiments after normalization against their respective *ACT1* signal and uncut T0 sample is shown.

## Genome-wide analysis of DNA repair in the absence of Cdc14 activity retrieved over-resection of the HO-induced break

To get insight into the specific role of Cdc14 in the repair of DSBs by HR, we investigated whether the DNA repair defects observed in the absence of the phosphatase could be due to a possible role of Cdc14 in controlling end resection dynamics. We took samples under the same experimental conditions described above and performed Southern blots using probes that hybridize at increasing distances from the HO-induced break (Fig. 2a). Resection progression, as measured by the time needed to reduce the band signals between 0 and 4 h after HO induction for each probe assayed, was similar between the wild-type and the *cdc14-1* mutant (Fig. 2b), indicating that Cdc14 is not required to trigger DNA end resection. Supporting this observation, both wild-type and *cdc14-1* cells showed similar resection kinetics in a non-reparable JKM179 background system (Supp. Fig. 1d). Moreover, Mre11 foci formation and dissolution, measured as a readout of DSB detection and resection initiation, produced similar values in both strains (Supp. Fig. 1e). The appearance of extended D-loop intermediates was also similar between the wild-type and the *cdc14-1* mutant, suggesting that strand invasion is perfectly accomplished in the absence of the phosphatase (Supp. Fig. 1f, g).

Four hours after HO induction, the wild-type strain showed a progressive restoration of all the bands, reaching a maximum intensity by 8 h after DSB induction. This kinetic was nearly identical to that previously reported using a similar inter-chromosomal DNA repair system[41]. However, in the absence of Cdc14 activity the accumulation of all DNA repair outcomes was compromised (Fig. 2b). Importantly, while wild-type cells showed only a small reduction in the disappearance of the distal 21 kb band, *cdc14-1* cells displayed a progressive decline in the intensity of this band throughout the entire experiment (Fig. 2b), suggesting that the phosphatase is required to avoid the accumulation of excessive long ssDNA during resection. Over-resection in *cdc14-1* cells is also observed when inducing an HO break at the *LEU2 locus* in the YMV80 background (Fig. 2c, d), a strain that uses break induced replication (BIR) and single strand annealing (SSA) for its repair[42], extending the role of Cdc14 in controlling resection length to other genomic locations and recombinational DNA repair systems.

The PMV strain contains a set of polymorphisms incorporated in the donor sequence at chromosome V that allows analyzing DNA repair in a genome wide sequencing approach[40]. Thus, we took advantage of this system to analyze at nucleotide resolution the effect of inhibiting Cdc14 activity during the formation of an HO-induced break. HO was expressed in exponentially growing wild-type and *cdc14-1* cells by adding galactose to the media, and we took samples at different intervals to perform genome-wide sequencing following the Illumina protocol, so that after random fragmentation of genomic DNA, only double-stranded fragments are incorporated into the sequencing libraries. Alignment of the sequencing reads to the reference genome

generated the coverage profiles shown in Supp. Fig. 2a. By normalizing the data of these coverage profiles against their respective undamaged T0 sample, we generated two-dimensional (2D) (Fig. 3a), three-dimensional (3D) (Supp. Fig. 2b), and colormap charts (Fig. 3b) representing the evolution of the HO-induced break over time.

To quantitatively measure the efficiency of resection in the absence of Cdc14 activity, we measured the efficiency of short- and long-range resection[40]. The short-range resection rate and symmetry, measured as the decline in coverage at the regions flanking the HO cleavage site by 1 h, was similar between wild-type and *cdc14-1* cells, with a slight preference to resect toward the left flank of the HO-induced break in both strains (Fig. 3c), indicating that the phosphatase is not required for the execution of this initial step of the repair pathway. Quantitative analysis of long-range resection was attained by calculating the correlation between the time and the distance needed to reduce the coverage from 1 to 0.5 to the left and right sides of the HO-induced break relative to the 0 h sample (Supp. Fig. 2c). As in wild-type cells, long-range resection in *cdc14-1* mutant was symmetrical and proceeded at a similar rate (Fig. 3d), confirming that the phosphatase is not involved in sustaining long-range resection.

In a wild-type strain, there were low levels of coverage surrounding the HO-induced break 4 h after HO induction. Subsequently, there was a gradual coverage recovery, with a maximum reached by 9-10 h (Fig. 3a, b and Supp. Fig. 2b). However, we detected a drastic reduction in coverage levels in the absence of Cdc14 activity when compared to the control strain, that progressively extended away from the HO-induced break along the time (Fig. 3a, b and Supp. Fig. 2b). Resection to the left side of the HO-induced break was blocked at a Ty1-LTR element (YCRWdelta11) located 32 kb from the DSB site (Fig. 3b, e), as previously reported for other DNA repair mutants that develop over-resection[40]. Resection to the right side of the HO break in *cdc14-1* cells extended up to the sub-telomere regions located at 100 kb from the DSB (Fig. 3b, e). It is important to remark that real-time polymerase chain reaction (qPCR) analysis of *cdc14-1* cells using oligonucleotides located -1.5 kb from the HO cleavage site retrieved values below 50% (expected for ssDNA) 12 h after HO induction (Fig. 3f), indicating that the over-resection observed in cells lacking Cdc14 activity affects ssDNA stabilization.

Overall, these results demonstrate that Cdc14 is essential for neither resection initiation nor progression of the resection machinery in response to an HO-induced break, but it is required to inhibit resection to avoid the formation of excessively long ssDNA.

## Cdc14-dependent resection inhibition relies on the exonuclease Dna2

We have shown above that lack of the phosphatase activity renders cells susceptible to over-resect the two sides of a repairable HO-induced break, indicating that Cdc14 is required to limit the extent of

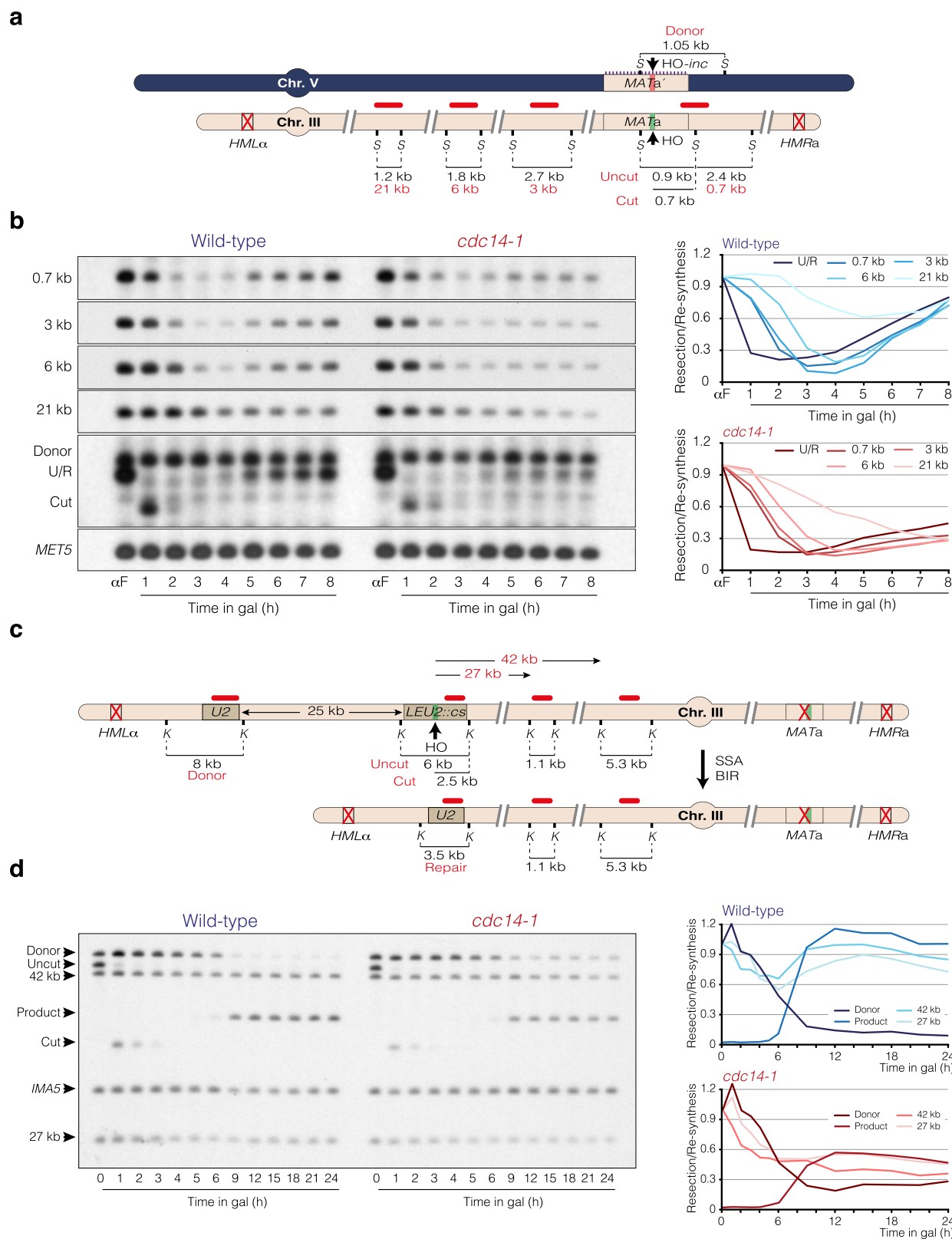

resection. Thus, we sought to investigate whether Cdc14 is involved in the negative regulation of the Exo1 or Sgs1-Dna2 resection pathways. We speculated that over-resection in *cdc14-1* cells should be bypassed when disrupting the exonuclease controlled by the phosphatase. We induced exponentially growing cultures of wild-type and *cdc14-1* cells by adding galactose to the media, transferred them to 33 °C, and took samples at different time intervals up to 24 h to perform Southern

blotting. As in pre-synchronized G1 cultures, resection in the wild-type strain took place between 1 and 4 h (Fig. 4a left panel). The farther the distance from the HO-induced break, the longer the time needed to reduce the intensities of the bands. Supporting the genome-wide analysis, we did not notice a drastic reduction in the 21 kb band, indicating that in most cells resection does not reach this distance from the HO-induced break. Four hours after DSB induction, we detected a clear

**Fig. 2 | Lack of Cdc14 produces over-resection of DSBs. a** Schematic representation of the DNA resection assay used to determine the extent of resection in the PMV background. The size of the bands for the "Uncut", "Cut", "Donor", and the different distances analyzed from the HO cleavage site are shown. The probes used to determine the kinetics of these bands through the repair of the HO-induced break are depicted with horizontal red lines. *S*: *Sty*I. **b** Southern blot analysis of wild-type and *cdc14-1* PMV cells using the experimental approach shown in **a**. After overnight culture in YP-Raffinose at 25 °C, alpha factor (αF) was added to block the cells in G1. After arrest, cells were released by removing the pheromone, induced with galactose, and transferred to 33 °C. Samples were taken at different intervals after HO induction, and genomic DNA was extracted, digested with *Sty*I and analyzed by Southern blot using the probes depicted in **a**. A *MET5* gene sequence was used as a loading control. The graphs on the right represent the quantification of the averaged band signals from two Southern blots after normalization against their respective *MET5* signal and uncut T0 sample. Note that in both strains, proximal probes retrieved to a higher level before that of the distal probes from the HO-induced break, a feature that might account for DNA repair by SDSA. U/R:

Uncut/Repair. **c** Schematic representation depicting relevant genomic structures of the YMV80 background used to assess DNA repair by SSA/BIR. The location of the HO cleavage site at the *LEU2 locus* (*LEU2::cs*) and the donor sequence (*U2*) are depicted. The size of the bands for the "Donor", "Uncut", "Cut", "Repair" and the different distances used to assess resection dynamics (27 kb and 42 kb to the right of the HO site) are shown. The location of the probes (horizontal red lines) and the restriction endonuclease cleavage sites (*K*: *Kpn*I) used for Southern blot analysis are illustrated. **d** Southern blot analysis of wild-type and *cdc14-1* YMV80 cells using the experimental approach shown in **c**. After overnight culture in YP-Raffinose at 25 °C, cells were induced with galactose and transferred to 33 °C. Samples were taken at different intervals after HO induction, genomic DNA was extracted, digested with *Kpn*I and analyzed by Southern blot using the probes depicted in **c**. A *IMA5* gene sequence was used as a loading control. The graphs on the right represent the quantification of the averaged band signals from two Southern blots after normalization against their respective *IMA5* signal and uncut T0 sample. Quantification of the product formation was attained by normalizing against the *IMA5* and the T24 sample of the wild-type strain.

re-accumulation of all bands tested, reflecting continuous re-polymerization of the resected DNA. Analysis of *cdc14-1* cells under these experimental conditions corroborates continuous over-resection spanning from 1 to 24 h, reaching the farthest distance analyzed at 27 kb (Fig. 4a right panel). It is important to note that Cdc14-dependent resection inhibition is not due to the presence of *MATa'* polymorphisms in the PMV background strain because the insertion of a non-polymorphic donor sequence in chromosome V also renders cells to an over-resection phenotype in the absence of Cdc14 activity (Supp. Figure 3a).

Next, we asked whether the absence of Exo1 might constrain over-resection of *cdc14-1* cells under the same experimental conditions used above. An *exo1Δ* mutant displayed a reduction in the disappearance of the 3 and 6 kb bands (Fig. 4b left panel) due to the role of this exo-nuclease to sustain resection[4,5] (Supp. Figure 3b). However, lack of Exo1 on a *cdc14-1* background displayed a similar over-resection phenotype to that observed in a simple *cdc14-1* mutant (Fig. 4b right panel), indicating that Exo1 does not account for the over-resection observed in the absence of the phosphatase activity. Elimination of the resection inducer Fun30 also did not suppress over-resection of *cdc14-1* cells, confirming the existence of distinctive targets of the phosphatase in resection inhibition (Supp. Figure 3c).

To determine the possible contribution of the Sgs1-Dna2 pathway in the over-resection of *cdc14-1* cells, and since Dna2 is essential for the viability in *S. cerevisiae*, we generated an auxin-inducible degron of Dna2 to rapidly induce degradation of the protein when adding auxin to the media. We added galactose to exponentially growing cells to induce the HO break and transferred the culture to 33 °C in the absence of auxin for 6 h so as to not interfere with resection progression during the initial stages of the response. Six hours after HO induction, we split the culture into two and added auxin or mock ethanol. Both cultures behaved similarly in FACS analysis, with cells entering in the DNA damage checkpoint arrest 3 h after galactose addition, and remaining blocked with 2c content for the entire experiment, due to the intrinsic anaphase arrest inherent of the *cdc14-1* mutant (Supp. Figure 4a). Analysis of Dna2 stability by western blot confirmed that, while the addition of mock ethanol did not affect the stability of the protein, the addition of auxin to the media led to its rapid depletion (Supp. Figure 4b top panel). Confirming the reduction in Dna2 activity, we detected high levels of Rad53 phosphorylation after adding auxin to the culture (Supp. Figure 4b bottom panel), consistent with previous report[43]. Southern blot assays under these experimental conditions demonstrated that while the addition of mock ethanol resulted in an over-resection phenotype similar to that observed in *cdc14-1* cells (Fig. 4c left panel), the addition of auxin to the media impeded the disappearance of the long 21 and 27 kb band signals and facilitated the re-accumulation of the band signals closer to

the HO-induced break (Fig. 4c right panel). This effect is even more noticeable when comparing *cdc14-1* and *cdc14-1* Dna2-AID strains (Supp. Figure 4c), suggesting that the AID tag might exert a negative effect on the nuclease activity of Dna2. Resection by Dna2 relies on the helicase activity of Sgs1[5,44,45]. However, lack of Sgs1 did not rescue over-resection of *cdc14-1* cells (Supp. Figure 4d), probably due to the marked over-resection phenotype inherent of *sgs1Δ* cells in the PMV background (Supp. Figure 4d left panel and[40]).

Taken together, these results corroborate that Dna2 is the exo-nuclease responsible for the over-resection observed in *cdc14-1* cells and suggest that Cdc14 contributes to inhibit resection by negatively regulating Dna2 activity.

## Cdc14 inhibits resection by targeting Cdk-dependent phosphorylation on Dna2

Dna2 is phosphorylated by the Cdk on residues S17 and S237 to enhance its nuclear localization and consequently, its recruitment to DSBs[11]. Besides, Cdc14 is the only reported phosphatase capable of counteracting Cdk-dependent phosphorylation[14]. Thus, we wondered whether the steady state phosphorylation of these residues by Cdk/Cdc14 during the damage response could modulate the extent of resection. To test this hypothesis, we expressed under their own pro-moters a wild-type Dna2, a phospho-deficient Dna2[S17-237A], and a phospho-mimetic Dna2[S17-237D] versions on the *cdc14-1* Dna2-AID PMV background and checked for the length of resection when expressing these nuclease variants.

First, we induced the HO break in a *cdc14-1* Dna2-AID expressing a wild-type version of Dna2 by adding galactose to the media, transfer-ring the culture to 33 °C, and 6 h later adding auxin to induce degra-dation of the endogenous Dna2. The addition of auxin to the *cdc14-1* Dna2-AID partially recovered the capacity of cells to promote DNA re-synthesis after resection (Fig. 5a left panel). However, the expression of wild-type Dna2 on this background sustained resection up to 24 h after the HO-induced break for all analyzed bands (Fig. 5a right panel), in a manner similar to that in the single *cdc14-1* mutant (Fig. 4a right panel). This result confirms that the exogenously expressed Dna2 protein is completely functional and can be used for the subsequent testing of the phospho-Dna2 variants.

Next, we reasoned that if the constitutively phosphorylation of Dna2 on S17 and S237 in the absence of Cdc14 activity is behind the detected over-resection, a phospho-deficient version of these resi-dues should reduce the length of resection in *cdc14-1* cells. Thus, we expressed a phospho-deficient Dna2[S17-237A] on the same *cdc14-1* Dna2-AID background and tested for its resection capacity. As predicted, the expression of a phospho-deficient Dna2[S17-237A] reduced the over-resection of *cdc14-1* cells (Fig. 5b), suggesting that Cdc14 counteracts Cdk phosphorylation on residues S17 and S237 of

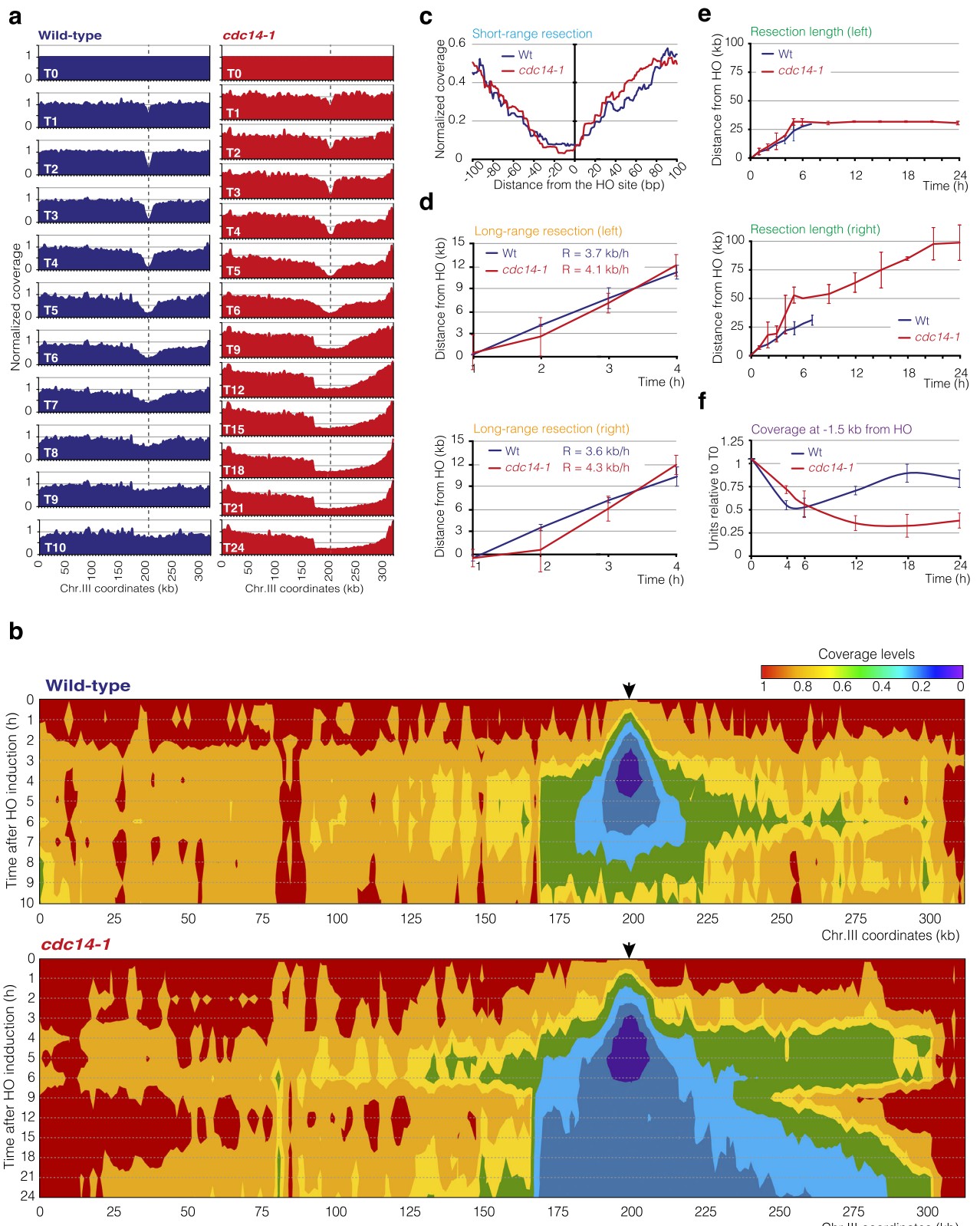

Dna2 to inhibit its activity, thus limiting the extent of resection. To corroborate this hypothesis, we also tested for the extent of resection after expressing a phospho-mimetic Dna2$^{S17-237D}$ version on the *cdc14-1* Dna2-AID background. Expression of this phospho-mimetic version of Dna2 displayed an over-resection similar to that observed when expressing wild-type Dna2 (Fig. 5c). The over-resection capacity of the *cdc14-1* Dna2-AID Dna2$^{S17-237D}$ strain was better perceived when comparing it to the *cdc14-1* Dna2-AID strain (Supp. Figure 5a).

Taken together, these results suggest that Cdc14 controls the extent of resection by counteracting Cdk-dependent phosphorylation of Dna2 at S17 and S237.

**Fig. 3 | Genome-wide sequencing analysis of the HO repair in wild-type and *cdc14-1* PMV strains. a** Normalized 2D coverage profiles of wild-type and *cdc14-1* PMV cells. Vertical dotted lines mark the position of the HO cleavage site. The graph shows the mean from two biologically independent experiments. **b** Normalized 2D coverage profiles from **a** were compiled to generate colormap charts of wild-type and *cdc14-1* PMV strains. These graphs represent simultaneous coverage levels (defined by the red, green, and blue colors) normalized against the T0 sample, the coordinates along chromosome III (*x*-axis), and the time after HO induction (*y*-axis). Black arrows mark the position of the HO cleavage site. The graphs represent the mean from two biologically independent experiments. **c** Analysis of short-range resection in wild-type and *cdc14-1* PMV cells. The graph shows the mean from two biologically independent experiments. **d** Analysis of left and right long-range resection (top and bottom respectively) in wild-type and *cdc14-1* PMV cells. The long-range resection rate (R) is shown. The graphs show the mean ± SD from two biologically independent experiments. **e** Estimation of resection length to the left (top) and to the right (bottom) of the HO site in wild-type and *cdc14-1* PMV cells. The graphs represent the mean ± SD from two biologically independent experiments. **f** Real time PCR of samples taken at different intervals to assess ssDNA stabilization after HO expression in wild-type and *cdc14-1* PMV cells. The coverage levels were determined by using a pair of oligonucleotides located at 1.5 kb to the left side of the HO-induced break. Amplification with a pair of oligonucleotides located at the *ACT1* gene was used for DNA content normalization. The values were normalized against the T0 sample. The graph shows the mean ± SD from two biologically independent experiments.

## Cdc14 inhibits resection by Dna2 mainly during its mitotic activation

In non-DNA damage conditions, the early and late anaphase exclusion of Cdc14 from the nucleolus (FEAR and MEN, respectively) activates the phosphatase during mitosis to promote Cdk inactivation and mitotic exit[32]. Consequently, *cdc14-1* mutants arrest cell cycle progression in late anaphase[14]. However, Cdc14 can also be transiently released from the nucleolus during the G2 arrest induced by the DNA damage checkpoint in response to DNA damage[35]. Therefore, we wondered whether the nucleoplasmic accumulation of Cdc14 occurring in response to a DNA lesion is sufficient to restrain resection progression. To test this possibility, we induced an HO break in exponentially growing wild-type and *cdc14-1* cells, transferred them to 33 °C, and added nocodazole after 6 h from the HO induction to arrest cells in metaphase after repair, thus avoiding the activation of mitotic Cdc14. Under these conditions only DNA damage-dependent nucleolar liberation of Cdc14 takes place (Fig. 6a). In parallel, we performed the same experiment but added the pheromone alpha factor to allow progression to the next G1 phase, thus permitting mitotic Cdc14 activation (Fig. 6a). We took samples every hour from the HO break induction to determine the dynamics of resection by Southern blot. Wild-type cells displayed moderate over-resection in cultures treated with nocodazole but not when alpha factor was added (Fig. 6b top panel). This result is in line with recent discoveries showing that nocodazole treatment impairs DNA repair by HR[46]. Besides, a *cdc14-1* mutant showed marked over-resection independently of the treatment with nocodazole or alpha factor (Fig. 6b bottom panel). These results indicate that, although the accumulation of Cdc14 at the nucleoplasm during the damage response slightly contributes to resection inhibition, the phosphatase performs this function mainly during its mitotic activation.

It has been shown that Cdk-dependent Dna2 phosphorylation of S17 and S237 stimulates its recruitment to DSBs[11]. Therefore, we wondered whether Cdc14 inhibits resection by removing Dna2 from the DNA lesion. To test this hypothesis, we induced the HO break in exponentially growing wild-type and *cdc14-1* cells containing GFP-tagged Dna2 expressed under its own promoter, transferred cells to 33 °C, and 6 h later added nocodazole or alpha factor to the media. We took samples every 2 h from HO induction to determine the percentage of cells with Dna2 foci. Both wild-type and *cdc14-1* backgrounds showed 80% of Dna2 foci formation 4 h after inducing the HO break, corroborating the ability of *cdc14-1* cells to initiate resection. In agreement with the over-resection of wild-type cells treated with nocodazole after the HO-induced break (Fig. 6b left panel), we only detected a modest reduction in the number of Dna2 foci during the experiment (Fig. 6c middle-top panel and top graph) and a transient accumulation of Cdc14 in the nucleoplasm by 6 h after the HO induction (Supp. Figure 5b left graph) during the G2/M arrest (Supp. Figure 5c). However, after inducing the HO break, wild-type cells treated with alpha factor progressed to the subsequent G1 phase (Supp. Figure 5c), thus increasing the number of cells with nucleoplasmic Cdc14 during the experiment (Supp. Figure 5b right graph) due to the

activation of the FEAR/MEN pathways. Under these conditions we detected a drastic reduction in the percentage of Dna2 foci (Fig. 6c right-top panel and bottom graph), corroborating the ability of these cells to inhibit resection. The absence of Cdc14 activity completely impaired the resolution of Dna2 foci in the presence of either nocodazole or alpha factor throughout the entire experiment (Fig. 6c middle and right-bottom panels and graphs). These results indicate that mitotically activated Cdc14 is the main responsible for the removal of Dna2 foci at the DSBs to ensure resection inhibition.

Next, we wondered whether Cdc14 modulates Dna2 function by regulating its steady state phosphorylation during the damage response. In a first approach, we inserted at the *TRP1 locus* a Cdc14 tagged with the GFP fusion regulated under the galactose promoter in a *cdc14-1* PMV background. We induced both Cdc14 and HO expression by adding galactose to the media and transferred cells to 33 °C to inactivate the endogenous *cdc14-1* allele. After 1.5 h, we added glucose to the culture to repress the expression of Cdc14 and collected samples every hour to follow Cdc14 localization. Cdc14 over-expression leads to its accumulation at the nucleoplasm even at early time points after the HO-induced break (Fig. 7a top panel). Interestingly, Dna2 was rapidly dephosphorylated 1 h after the addition of galactose, indicating that over-expressed Cdc14 induces Dna2 dephosphorylation (Fig. 7b top panel). This effect is not due to the possible inhibition of the Cdk generated by the excessive accumulation of Cdc14, because the B subunit of the Polα-primase complex (Pol12), used as reporter of Cdk activity, was efficiently phosphorylated under these experimental conditions (Fig. 7b middle panel). Moreover, Rad53 was phosphorylated in a timely manner (Fig. 7b bottom panel) and cells were efficiently blocked at the G2/M damage checkpoint, as denoted by the appearance of dumbbell cells in the culture (Fig. 7a bottom panel).

To evaluate the capacity of native Cdc14 to target Dna2 for dephosphorylation, we followed the phosphorylation of endogenous Dna2 in wild-type and *cdc14-1* PMV cells after the induction of an HO break. We pre-synchronized both strains in G1 by adding alpha factor to enrich the cultures in cells with unphosphorylated Dna2 and released cells into fresh media in the presence of galactose to induce the HO break. Western blot analysis revealed the appearance of high-molecular-weight Dna2 from 1.5 h after the galactose induction. After 6 h, we detected the accumulation of low-molecular-weight Dna2, indicating that the nuclease was actively being dephosphorylated from this time point (Fig. 7c top panel). However, lack of Cdc14 endorses the accumulation of high-molecular-weight forms of Dna2 during the entire experiment (Fig. 7c top panel). In agreement with the low levels of DNA repair observed in the *cdc14-1* mutant, Rad53 was hyper-phosphorylated in the absence of the phosphatase during the entire experiment (Fig. 7c bottom panel). In accordance with the role of mitotically activated Cdc14 to attain resection inhibition, we detected the appearance of dephosphorylated forms of Dna2 after inducing the HO break and adding alpha factor, but not when cells were blocked in metaphase by adding nocodazole (Fig. 7d left panels). In the absence of Cdc14, we detected high levels of Dna2 phosphorylation independently of the treatment with alpha factor or nocodazole (Fig. 7d, right

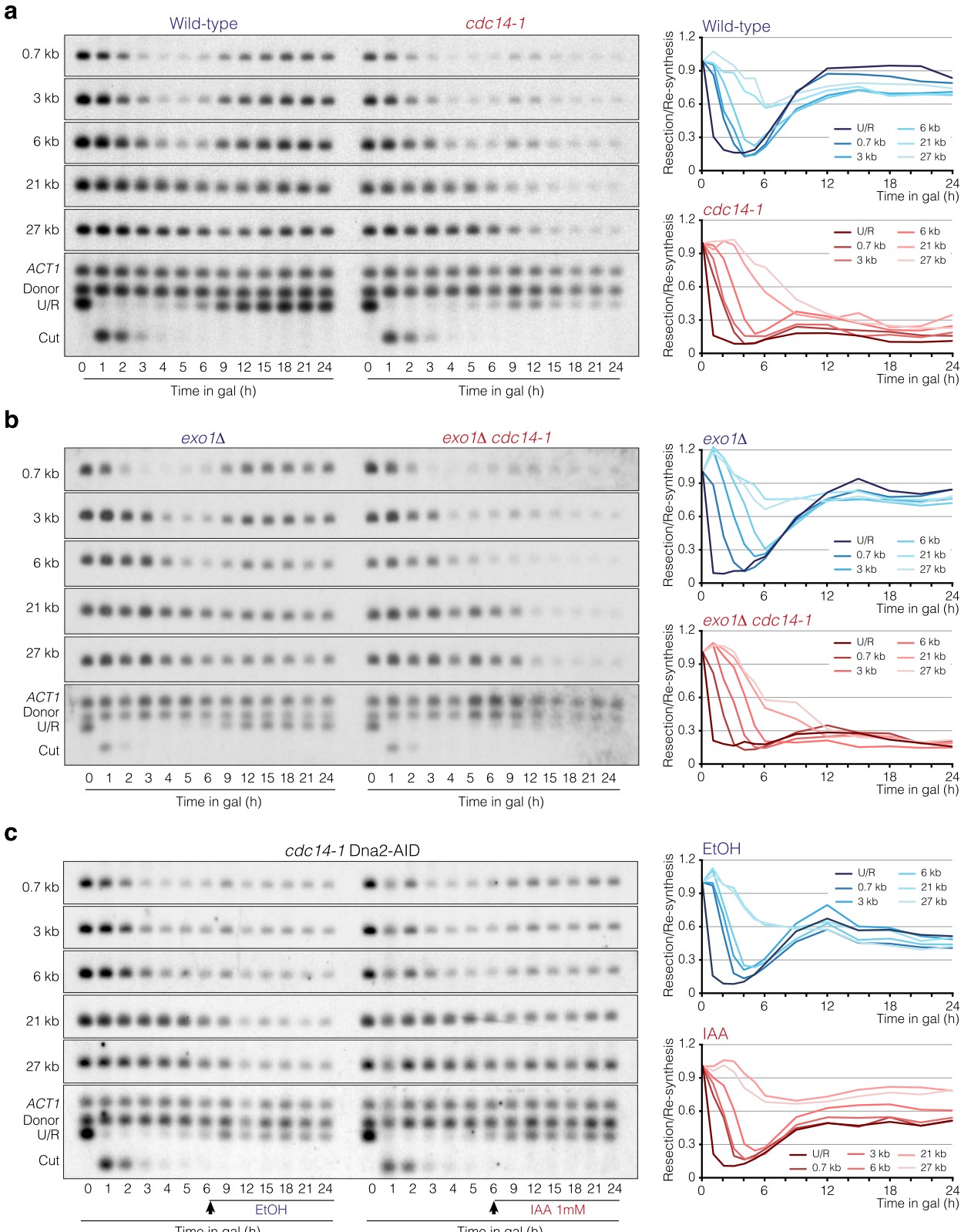

panels). This effect is better visualized when using a truncated Dna2$_{1-286}$ version containing most of the Cdk consensus sites (Fig. 7e, f).

It has been previously shown that Cdk phosphorylates Dna2 during an unperturbed cell cycle to import the nuclease into the nuclear compartment[10]. Accordingly, only the block in nocodazole (metaphase), but not in alpha factor (G1), promoted the nuclear

accumulation of the nuclease in the PMV strain growing in the absence of HO expression (Fig. 7g). Moreover, *cdc14-1* cells grown at restrictive temperature accumulate Dna2 in the nucleoplasm in non-DNA damage conditions (Fig. 7g), suggesting that Cdc14 might also target the nuclease during mitotic exit in undamaged cells.

Overall, these data indicate that mitotically activated Cdc14 induces Dna2 dephosphorylation to facilitate its removal from the DSB

**Fig. 4 | Over-resection of *cdc14-1* cells depends on Dna2. a** Southern blot analysis of resection and DNA repair in wild-type and *cdc14-1* PMV cells. After overnight culture in YP-Raffinose at 25 °C, HO was induced with galactose and cultures were transferred to 33 °C. Samples were taken at different intervals, and genomic DNA was extracted, digested with *StyI*, and analyzed by Southern blot using probes located at increasing distances from the HO cleavage site as depicted in Fig. 2a. An *ACT1* probe was used as a loading control. The graphs on the right represent the quantification of the averaged band signals from two Southern blots after normalization against their respective *ACT1* signal and uncut T0 sample. U/R: Uncut/Repair. **b** Southern blot analysis of resection and DNA repair in *exo1Δ* and *exo1Δ cdc14-1* PMV cells. Southern blot was performed under the same experimental conditions described in **a**. The graphs on the right represent the quantification of the averaged band signals from two Southern blots after normalization against their respective *ACT1* signal and uncut T0 sample. U/R: Uncut/Repair. **c** Southern blot analysis of resection and DNA repair in *cdc14-1* PMV cells carrying an inducible auxin degron of Dna2. After overnight culture in YP-Raffinose at 25 °C, HO was induced with galactose and cells were transferred to 33 °C. Six hours after the HO induction the culture was split, and mock ethanol or auxin was added. Samples were taken before and after the addition of EtOH/auxin and treated as in **a**. The graphs on the right represent the quantification of the averaged band signals from two Southern blots after normalization against their respective *ACT1* signal and uncut T0 sample. U/R: Uncut/Repair.

boundary and to facilitate resection inhibition, thus avoiding the generation of excessively long ssDNA molecules that might be incompatible with the proficient repair of DSBs by HR.

## Lack of Cdc14 affects DNA re-synthesis

Because Cdc14 is required to inhibit resection by promoting Dna2 dephosphorylation, we focused on determining the consequences of the over-resection on the efficiency of DNA repair by HR. An advantage of the PMV strain is that it can be used to measure the symmetry and kinetics of DNA re-synthesis by determining the time after resection needed to recover half of the coverage at any given distance from the HO-induced break[40]. To check for a role of Cdc14 in regulating DNA re-polymerization, we calculated the relative DNA re-synthesis rates at 0.5 and 15 kb to the left and right sides of the HO-induced break by using the trend line equations obtained from the coverage profiles of wild-type and *cdc14-1* cells (Supp. Figure 6a). In wild-type cells, DNA re-synthesis measured at 0.5 kb from the DSB displayed a uniform and symmetrical coverage re-polymerization rate along the entire experiment, with a DNA re-synthesis initiation rate of 0.12 and 0.13 to the left and right of the HO-induced break, respectively (Fig. 8a and Supp. Figure 6b). The percentage of cells reaching a distance of 0.5 kb to the left and right side of the HO-induced break was 74% and 82%, respectively (Fig. 8a). By contrast, cells lacking Cdc14 activity showed a drastic reduction in the DNA re-synthesis rate to 0.07 measured between 4 and 9 h in both directions (Fig. 8a and Supp. Figure 6b). Moreover, the percentage of cells that reached 0.5 kb to the left and right side of the HO-induced break was reduced to 35% and 37%, respectively, at T9, a time point where coverage recovery was blocked until the end of the experiment (Fig. 8a). These data indicate that Cdc14 is required to promote DNA re-polymerization on both sides of the HO-induced break. Similar results were observed when measuring DNA re-synthesis at 15 kb from the HO-induced break, indicating that the defects in DNA re-polymerization inherent of *cdc14-1* cells are extended to distal regions of the HO cleavage site (Fig. 8a). These results suggest that the over-resection of DSBs observed in the absence of Cdc14 activity might affect the re-polymerization of the DNA molecule during the recombinational repair pathway.

## A genome-wide analysis of discordant read pairs reveals a role for Cdc14 in the distribution of GC events

We have seen that cells lacking Cdc14 activity are compromised in DNA re-synthesis. Thus, we wondered whether this effect might affect the distribution of GC events at the break site after repair. The PMV strain harbors a 1.3 kb long *MATa'* fragment in chromosome V containing 23 point mutations each separated by 60 nt (Fig. 1a), making it possible to analyze the length and directionality of GC by measuring the increase in *MATa'* polymorphisms at T10 relative to T0[40]. The normalized read coverage profile in the 1.6 kb region next to the HO/HO-*inc* sites at T10 revealed a gradual decay in the *MATa* coverage and an enrichment in the reads mapping to the *MATa'* to the left side of the HO-induced break (Supp. Figure 6c left panel). In accordance with the defects in DNA re-synthesis of *cdc14-1* cells mentioned above, lack of Cdc14 activity led to low levels of coverage recovery at the entire *MATa locus* in the T24

sample relative to the T0 sample, defects that were exacerbated to the left side of the HO-induced break (Supp. Figure 6c right panel). We also detected a slight accumulation of *MATa'* reads at the left side of the HO-*inc*, indicating that some cells incorporated polymorphisms from chromosome V into chromosome III (Supp. Figure 6c right panel).

To measure GC between the *MATa* and *MATa'* loci, we determined the fraction of each polymorphism of *MATa'* relative to *MATa* and applied a correction factor to compensate for the lack of chromosome III repair at T10 for wild-type and T24 for *cdc14-1* samples, as reported previously[40]. While we did not observe accumulation of *MATa'* relative to *MATa* polymorphisms at T0 in either strain, we detected an accumulation of the HO-*inc* that decreased gradually to the left distal side of the HO-induced break at T10 and T24 in wild-type and *cdc14-1* cells, respectively (Fig. 8b). In accordance with the lack of DNA re-synthesis from 9 h after the HO induction in the absence of Cdc14, a similar profile was obtained when processing the T9 sample in the *cdc14-1* strain (Supp. Figure 6d). This result suggests that those cells capable of repairing the HO-induced break in the absence of Cdc14 do not show significant defects in the asymmetry of GC. However, because only 26% of the *cdc14-1* cells repaired the DNA lesion at T24 (Fig. 8a), the correction factor obtained for this analysis was moderately high, thus impeding accurate measurement of the GC distribution at each side of the break.

To sort this problem out, we decided to improve our GC analysis by using a different approach, that involved analyzing discordant read pairs obtained from the paired-end sequencing alignment of our genomic libraries. Briefly, a discordant read pair is defined as a pair of reads from the same DNA fragment whose alignments to the reference genome retrieve a distance and/or position that differ from the expected if the entire fragment was continuous on the reference genome. In our method, we used discordant read pairs to identify GC discontinuities at the *MATa-MATa'* boundary after the incorporation of polymorphisms from chromosome V into chromosome III during DNA repair (Supp. Figure 6e). Resembling the DNA re-polymerization profile (Fig. 8a), the proportion of discordant *MATa-MATa'* reads pairs in relation to the total number of reads analyzed in the wild-type strain accumulated progressively beginning 5 h after the DSB induction, reaching a maximum of $1.2 \times 10^{-5}$ by 10 h after HO expression (Supp. Figure 6f). In agreement with the DNA re-polymerization profile measured in *cdc14-1* cells (Fig. 8a), the appearance of discordant reads in cells lacking Cdc14 activity was delayed and reached a maximum of $0.7 \times 10^{-5}$ by 9 h after the HO induction (Supp. Figure 6f), confirming the reduced number of GC events in the absence of the phosphatase. Validating the specific accumulation of discordant read pairs between the recipient and donor chromosomes (Chr. III and Chr. V, respectively), fewer discordant read pairs were detected when assessing chromosome III with a chromosome not involved in the repair of the HO-break (i.e., Chr. VIII) in both wild-type and *cdc14-1* cells (Supp. Figure 6f).

To map the position of each individual recombination event precisely, we plotted the *MATa* (x-axis) and *MATa'* (y-axis) coordinates for each discordant read pair obtained from the entire set of samples (T0 to T10/T24) (Fig. 8c and Supp. Figure 6g). In these graphs, the HO-*inc* axis divides the incorporation of polymorphisms to the left

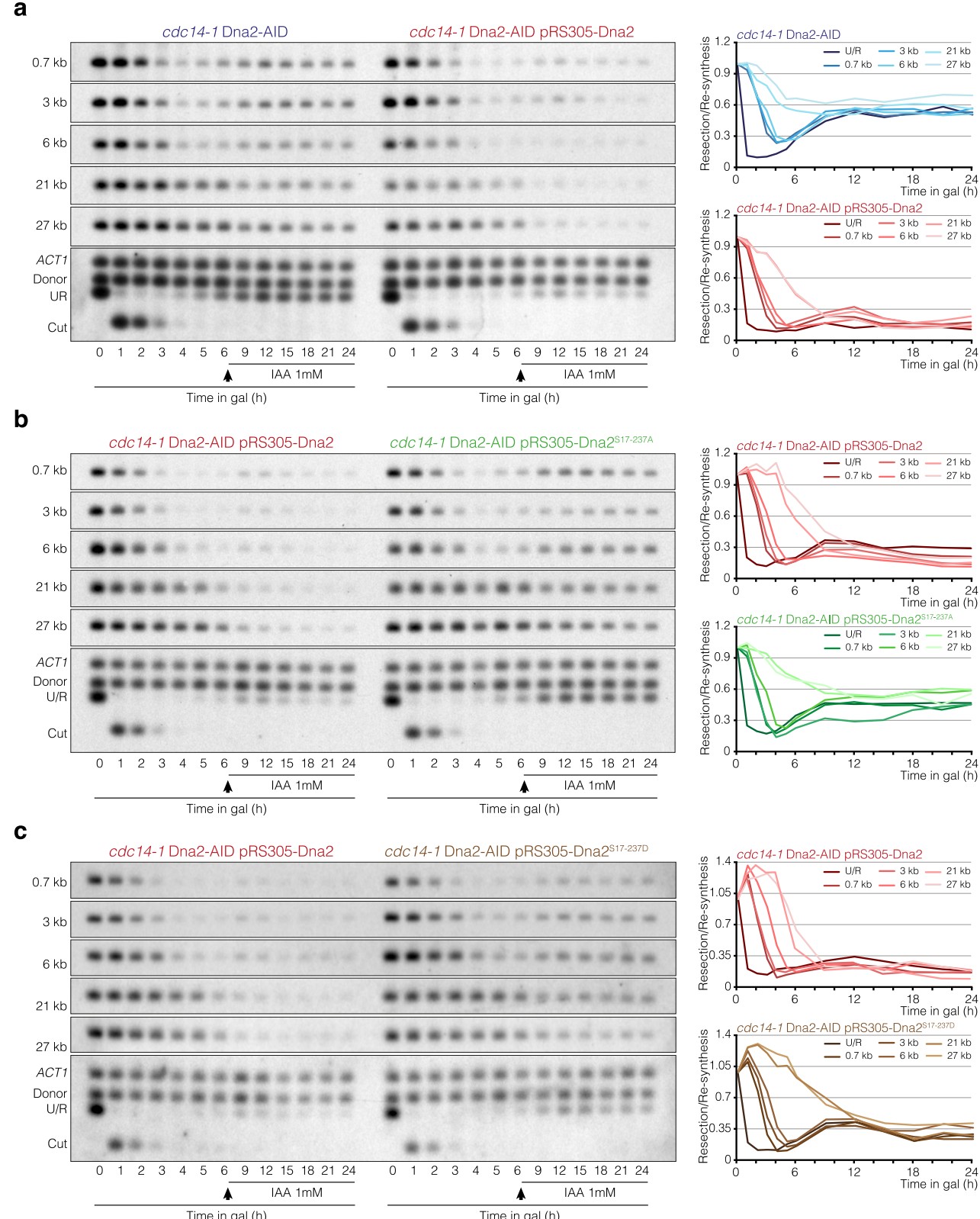

(bottom quadrants) from those occurring to the right (top quadrants) side of the HO-induced break (Fig. 8c). Discordant reads pairs at the left-bottom quadrant and right-bottom quadrant represent respectively, the left and right boundaries of GC tracts taking place to the left side of the HO break. Discordant reads pairs at the left-top quadrant and right-top quadrant represent respectively, the left and right boundaries of GC tracts occurring to the right side of the HO break. As

expected, most discordant read pairs followed a regression line, and were enriched at the bottom quadrants of the graph, indicating that most of the GC events take place to the left of the HO-induced break, confirming the asymmetry of GC in both wild-type and *cdc14-1* cells (Fig. 8c and Supp. Figure 6h).

The coverage analysis associated to the discordant read pairs showed the presence of two regions in chromosome III and V where

**Fig. 5 | Cdc14 inhibits resection by counteracting Cdk-dependent phosphorylation of Dna2. a** Southern blot analysis of resection and DNA repair in *cdc14-1* Dna2-AID PMV cells expressing a wild-type version of Dna2. After overnight culture in YP-Raffinose at 25 °C, HO was induced with galactose and the cultures were transferred to 33 °C. Six hours after HO induction, auxin was added to inactivate the endogenous Dna2-AID. Samples were taken before and after the addition of auxin, and genomic DNA was extracted, digested with *Sty*I, and analyzed by Southern blot using probes located at increasing distances from the HO cleavage site as depicted in Fig. 2a. An *ACT1* probe was used as a loading control. The graphs on the right represent the quantification of the averaged band signals from two Southern blots after normalization against their respective *ACT1* signal and uncut T0 sample. U/R: Uncut/Repair. **b** Southern blot analysis of resection and DNA repair in *cdc14-1* Dna2-AID PMV cells expressing a phospho-deficient version of Dna2. Southern blot was performed under the same experimental conditions as in **a**. The graphs on the right represent the quantification of the averaged band signals from two Southern blots after normalization against their respective *ACT1* signal and uncut T0 sample. U/R: Uncut/Repair. **c** Southern blot analysis of resection and DNA repair in *cdc14-1* Dna2-AID PMV cells expressing a phospho-mimetic version of Dna2. Southern blot was performed under the same experimental conditions as in **a**. The graphs on the right represent the quantification of the averaged band signals from two Southern blots after normalization against their respective *ACT1* signal and uncut T0 sample. U/R: Uncut/Repair.

the *MAT*a and *MAT*a' reads were enriched in both wild-type and *cdc14-1* mutant (blue and red coverage profiles in Fig. 8c), suggesting the presence of defined regions where GC takes place. Interestingly, because the distribution of discordant read pairs defines the two boundaries of the GC events (Supp. Figure 6e), we can precisely infer the directionality and extent of GC by determining the coordinates where the *MAT*a and *MAT*a' peaks intersect (see the *MAT*a and *MAT*a' discordant read pairs coverage profiles in blue and red, respectively, Fig. 8d). Both wild-type and *cdc14-1* cells presented a similar distribution of the *MAT*a-*MAT*a' intersecting peaks, suggesting that the symmetry of GC is maintained in the absence of Cdc14 activity (Fig. 8d). Still, and in agreement with the poor DNA repair efficiency observed in the absence of Cdc14, the coverage levels of discordant read pairs were reduced in the absence of the phosphatase (Fig. 8d and Supp. Figure 6f).

To deepen the insight obtained from this analysis, we quantitatively measured the distribution of the *MAT*a-*MAT*a' discordant read pairs after repair. Both wild-type and *cdc14-1* cells displayed similar levels in the proportion of discordant read pairs located at the right proximal (10.9% and 8.6%, respectively) and distal (5.7% and 6.5%, respectively) sides of the HO-induced break (D and E in Fig. 8d, e). The average length of the proximal right GC was similar for the wild-type and the *cdc14-1* mutant (95 nt and 99 nt, respectively) (Fig. 8f). This confirms that Cdc14 is not involved in the asymmetry of GC during HR repair. The percentage of discordant read pairs at the distal left flank of the HO-induced break in wild-type and *cdc14-1* cells was 31.6% and 22.2%, mapping at 616 and 619 bp, respectively, indicating that lack of Cdc14 affects GC extension to the left side of the HO-induced break (A in Fig. 8d, e, f). Accordingly, we identified an increase from 7.8% to 14.6% in the proportion of discordant read pairs located between the left proximal and distal region of the HO-induced break between wild-type and *cdc14-1* cells (B in Fig. 8d, e). Moreover, the number of discordant read pairs mapping to the left proximal flank of the HO cleavage site (C in Fig. 8d, e) rose from 44% in the wild-type to 48.1% in the *cdc14-1* mutant, confirming the accumulation of shorter GC tracts to the left side of the HO-induced break in the absence of the phosphatase. In line with this result, the reduction in the number of discordant read pairs detected in A correlated with the increase in those spotted in B-E in the absence of Cdc14 activity (Fig. 8e).

These results confirm that analysis of discordant read pairs is a powerful approach to determine the precise distribution of recombinational events during DNA repair and demonstrate that Cdc14-dependent resection inhibition is required to sustain GC extension to the left side of the HO-induced break during recombinational DNA repair.

## Discussion

In recent years, it has become evident that Cdk plays a role in the activation of multiple factors involved in the DNA damage response. One of the most relevant steps activated by Cdk during the recombinational DNA repair pathway is resection of broken DNA ends[6]. Several factors involved in this stage have been identified as Cdk targets. In response to DNA damage, the MRX subunits Mre11 and Xrs2 are

phosphorylated by the Cdk to activate resection, thus biasing DNA repair through HR[8]. Regulation of the MRX complex by phosphorylation seems to be an evolutionarily conserved mechanism, because the NBS1 subunit of the MRN complex (equivalent to MRX in budding yeast) is also phosphorylated in a Cdk-dependent manner in human cells to promote DNA repair by HR[47]. At the functional level, it has been proposed that the phosphorylation of MRX/MRN subunits by Cdk stimulates end-resection mainly by positively modulating Mre11 activity. Besides, it is well known that phosphorylation of other MRX-associated components is essential to promote a robust resection initiation. One example is Sae2, whose phosphorylation by Cdk is required for the assembly and disassembly of HR factors that stimulate resection[7,9]. Similar results have been obtained in humans, since Cdk-dependent phosphorylation of the Sae2 homolog CtIP induces the endonuclease activity of MRE11[48], mediates its interaction with BRCA1[49], facilitates its binding with NBS1[50], and stimulates DNA2 activity[51]. Importantly, Cdk also enhances resection by directly phosphorylating Dna2 during the formation of a DSB. Accordingly, substituting serine/threonine for alanine at the three Cdk consensus sites at the N-terminal domain of Dna2 (residues 4, 17, and 237) affects long-range resection. Because this mutant retains its nuclease activity and perfectly binds to other resection components[11], it seems that Cdk-dependent Dna2 phosphorylation might affect other properties of the nuclease. Supporting this view, Cdk enhances Dna2 nuclear accumulation[10] and recruitment to DSBs[11], suggesting a role for these Cdk phospho-sites in controlling resection by regulating the cellular distribution of the nuclease during the DNA damage response.

Although resection initiation and progression have been well studied in recent years, less is known about the molecular mechanisms that restrain resection once homology search and strand invasion has taken place. In humans, EXO1 phosphorylation by ATR marks it for SCF-mediated ubiquitination and degradation[52]. It has been proposed that activation and deactivation of EXO1 activity is regulated by the temporal control exerted by the tandem CDK/ATR during the DNA damage response. In this way, the accumulation of ssDNA by CDK-activated EXO1 might be sufficient to activate the ATM/ATR module, and thus its own inactivation. This feedback loop could constitute a biochemical switch to control the extent of resection. Supporting this hypothesis, Exo1 phosphorylation by Rad53 and Mec1 involves a negative feed-back loop that limits ssDNA accumulation during resection in the budding yeast[53]. Moreover, interaction of Exo1 with the 14-3-3 complex inhibits its damage recruitment to DNA ends, a mechanism that has been considered to prevent excessive accumulation of long ssDNAs[54–56].

We have shown in this work that in addition to Exo1, the Sgs1-Dna2 branch is also subjected to a temporary control that limits the extent of resection. Elimination of Cdc14 renders cells susceptible to a Dna2-dependent over-resection at both sides of the DSB, indicating that the phosphatase acts specifically on the Sgs1-Dna2 pathway to control resection length. Accordingly, accumulation of Cdc14 in the nucleoplasm during mitosis counteracts Dna2 phosphorylation, thus facilitating exclusion of the nuclease from the DSBs to block resection progression. Because Cdc14 is activated only when resection has been

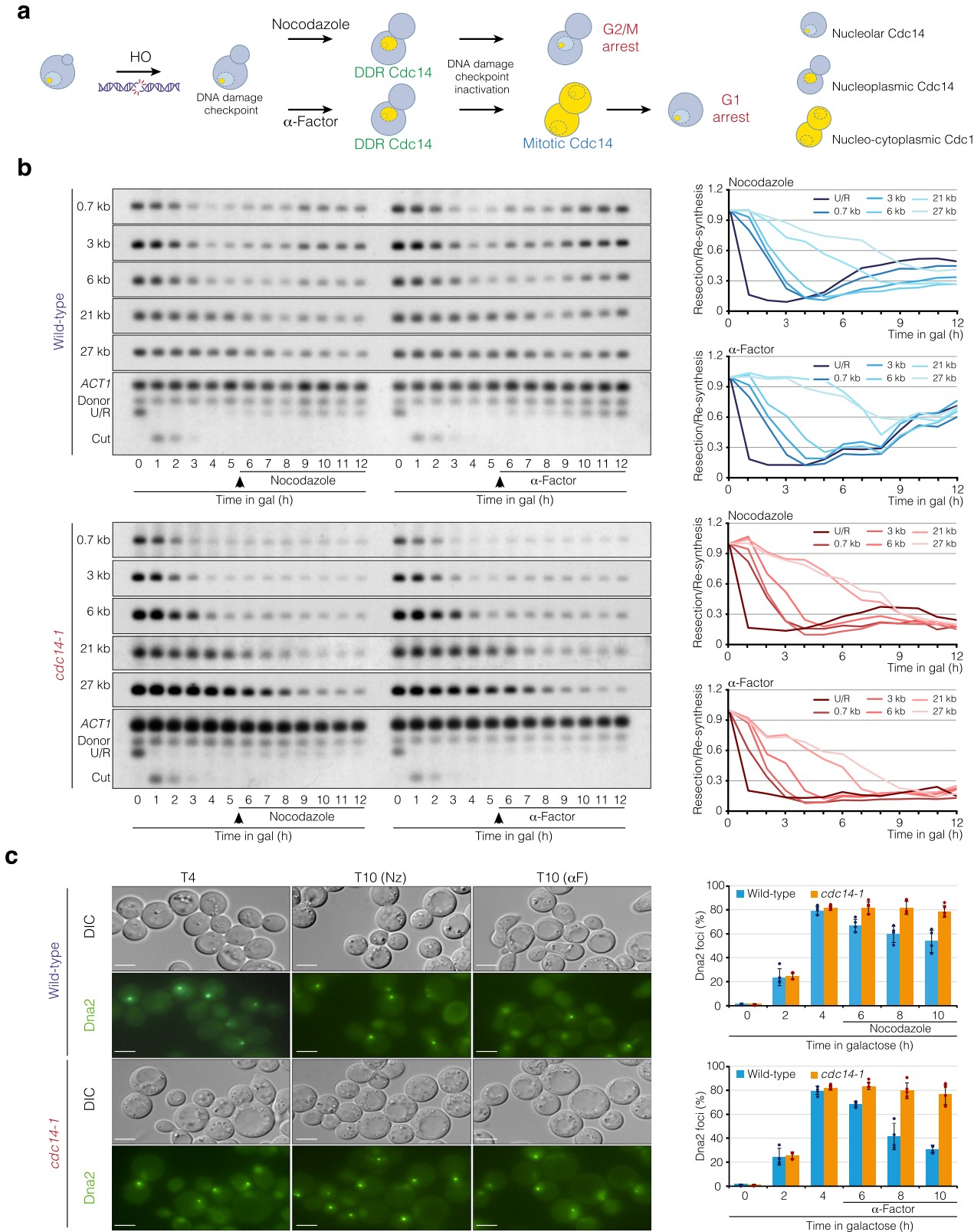

extended, it is intuitive to think that Cdc14 activity in response to a DNA lesion is subjected to a temporary control to avoid premature inhibition of resection during the early stages of the repair process, while efficiently blocking resection only when ssDNA tracts are long enough to support recombination. In this view, the sequestration of Cdc14 at the nucleolus during the initial steps of the DNA damage response would facilitate Cdk phosphorylation of targets required to

endorse a robust resection initiation. Interestingly, only mitotically activated Cdc14 inhibits resection, indicating that cells might enter mitosis with active resection. Taking into account that DNA re-synthesis and resection can coexist during repair of a DSB[40], and that the presence of ssDNA is the only prerequisite to activate the DNA damage checkpoint, it is possible that the re-polymerization of the resected tracts is sufficient to satisfy the damage checkpoint, thus

**Fig. 6 | Mitotic activated Cdc14 is the main responsible for resection inhibition.**
**a** Diagram representing the experimental approach used to determine the effect of DDR or mitotic Cdc14 in resection inhibition. In response to a HO break, cells activate the DNA damage checkpoint, thus inducing a G2/M arrest that facilitates the repair of the lesion. Addition of nocodazole at this stage maintains cells in G2/M during the entire experiment, a situation that only allows the liberation of DDR Cdc14. Alternatively, the treatment with alpha factor permits progression through mitosis upon DDR inactivation, thus triggering the activation of mitotic Cdc14 before blocking in the subsequent G1. Cdc14 is represented in yellow. DDR: DNA Damage Response. **b** Southern blot analysis using the experimental approach shown in **a** to determine the contribution of DDR/mitotic Cdc14 activation in resection inhibition. After overnight culture of wild-type (top panel) and *cdc14-1* (bottom panel) PMV cells in YP-Raffinose at 25 °C, HO was induced with galactose and cultures were transferred to 33 °C. Five-and-a-half hours after HO induction, nocodazole or alpha factor was added to the cultures. Samples along the entire

experiment were collected, and genomic DNA was extracted, digested with *Sty*I, and analyzed by Southern blot using probes located at increasing distances from the HO cleavage site as depicted in Fig. 2a. An *ACT1* probe was used as a loading control. The graphs on the right represent the quantification of the averaged band signals from three Southern blots after normalization against their respective *ACT1* signal and uncut T0 sample. U/R: Uncut/Repair. **c** Analysis of Dna2 foci kinetics using the same experimental approach shown in **a**. After overnight culture of wild-type (top panel) and *cdc14-1* (bottom panel) PMV cells containing a Dna2-GFP fusion regulated under its own promoter in YP-Raffinose at 25 °C, HO was induced with galactose and the cultures were transferred to 33 °C. Five-and-a-half hours after HO expression, nocodazole (Nz) or alpha factor (αF) was added. Samples were taken to visualize Dna2 foci formation/dissolution along the entire experiment. Representative maximum projection images at 4 h, 10 h (Nz) and 10 h (αF) from the HO induction are shown. Scale bar: 3 µm. The graphs represent the average percentage of Dna2 foci ± SD from three biologically independent experiments.

promoting mitotic entry with incomplete DSB repair. In accordance with this hypothesis, several lines of evidence indicate that DSBs continue to be processed via the recombinational repair pathway even once cells have entered mitosis[29,57–59]. It is important to remark that inter-chromosomal DNA repair in the PMV background is a slow process, in part due to the extensive resection at which DSBs are subjected[40], a situation that might differ from others HR repair systems. Moreover, ectopic recombination repair kinetics relies on multiple factors, such as the length of the donor sequence, the proximity between the recipient and the donor *loci* and the changes in chromatin structure and chromosome movements generated during DNA repair[60]. All these variables could affect the kinetics of inter-chromosomal DNA repair, including the extent of resection, the DNA repolymerization dynamics or the efficiency of DNA repair.

The negative regulation of Dna2 activity during the damage response might be evolutionarily conserved. In humans, DNA2 is inhibited by FANCD2[61], while in fission yeast, Pxd1 restrains resection by inhibiting the RPA-mediated activation of the 5′ endonuclease activity of Dna2[62]. If the CDK/CDC14 module could also constitute an additional mechanism to control DNA2 activity during the DNA damage response in high eukaryotes is a tantalizing question for the future. Interestingly, the role of CDK in stimulating resection is evolutionarily conserved and participates in phosphorylation of XBS1 and CtIP (orthologues of Xrs2 and Sae2, respectively) as well as EXO1[47,63,64] indicating that Cdc14-dependent resection inhibition might indeed be conserved throughout evolution.

Intriguingly, degradation of Dna2 does not fully restore the over-resection phenotype of *cdc14-1* mutant cells. This result, together with the fact that multiple components of the resection machinery are regulated by Cdk, suggests that Cdc14 might also contribute to resection inhibition by modulating other targets of the resection machinery. In agreement with this theory, multiple factors involved in DNA end resection have been shown to be regulated by Cdk phosphorylation, such as Mre11, Sae2, and Xrs2[6–9]. Additionally, resection inhibition has been proposed to be controlled independently of the biochemical regulation of resection factors. In this model, the replication machinery involved in DNA re-synthesis of the resected DNA catches up with the preceding processing nucleases to complete repair[65]. In this regard, the helicase Srs2 has been implicated in promoting resection inhibition by stimulating the removal of Rad51 from the nucleofilament, thus favoring polymerase loading and DNA re-synthesis. Because Srs2 has been shown to be regulated by the Cdk[13], is its tantalizing to propose that Cdc14 could also control Srs2 function during the DNA damage response to control resection length. Accordingly, we have shown that in the absence of Cdc14 cells develop a delay in re-synthesis initiation and a reduced net re-synthesis rate, similar phenotypes to those observed in the absence of Srs2[40]. Another possibility is that over-resection in *cdc14-1* cells might be a

consequence of alterations in other stages of the repair cycle. Since in the absence of Cdc14 activity we did not detect deficiencies in resection initiation or D-loop formation, these other putative roles of the phosphatase in DNA repair might be specifically constrained to the latest stages of the repair cycle, i.e., the resolution of recombination intermediates. Accordingly, the helicases Srs2 and Sgs1 and the regulatory subunit Mms4 of the resolvase Mus81 have been proposed to be controlled by the Cdk[13,66,67]. Whether Cdc14 participates in resection inhibition by modulating the steady-state phosphorylation of these factors is a crucial interrogation to address.

How might over-resection of DSBs impair cell viability in response to DNA damage? Timely inactivation of resection is essential for genome integrity because the accumulation of excessively long ssDNA might constitute a threat during DNA repair. This is due in part to the fragility of the ssDNAs molecules as well as to the risk to exhaust the pool of RPA[68,69]. In accordance with this view, we have shown that over-resection of DSBs in cells lacking Cdc14 destabilizes the long ssDNAs generated in the absence of the phosphatase, an effect that could interfere with the DNA re-synthesis step. Besides, extensive DNA end resection might also change the type of DNA repair used by the cell, from synthesis-dependent strand annealing (SDSA) to a more deleterious repair pathway, such as single strand annealing (SSA) or non-homologous end joining (NHEJ)[70]. In this regard, we have shown that lack of Cdc14 activity affects the frequency and distribution of GC events, indicating that extensive resection might alter the execution of the recombinational DNA repair pathway. Whether this observation is directly related to the role of Cdc14 in controlling the extent of resection or is due to its involvement in the regulation of other functions of the recombinational DNA repair pathway will be a decisive question for future studies.

## Methods
### Yeast strains and growing conditions
The strains used in this work are listed in Supp. Table 1. Strains JKM139 and YMV80 were kindly provided by J. Haber (Brandeis University, Waltham, MA, USA). Disruption and tagging of endogenous genes were achieved by gene targeting using PCR products as described previously[71,72] with oligonucleotides listed in Supp. Table 3. PMV and YMV80 derived strains[40] were induced by adding 2% galactose to cells growing in YP with 2% raffinose. Samples were collected for DNA analysis before and after adding galactose to the media. To inactivate the *cdc14-1* thermosensitive allele, cells growing at 25 °C were transferred to 33 °C. To deplete Dna2, we used the auxin-inducible degron (AID) system[73]. Auxin (Acros Organics, 70000010-3155), was used at 1 mM final concentration. Arrest in G1 was attained by using the pheromone alpha factor at 3 µM. Block in G2/M was performed by using the microtubule depolymerization drug nocodazole at 50 µM.

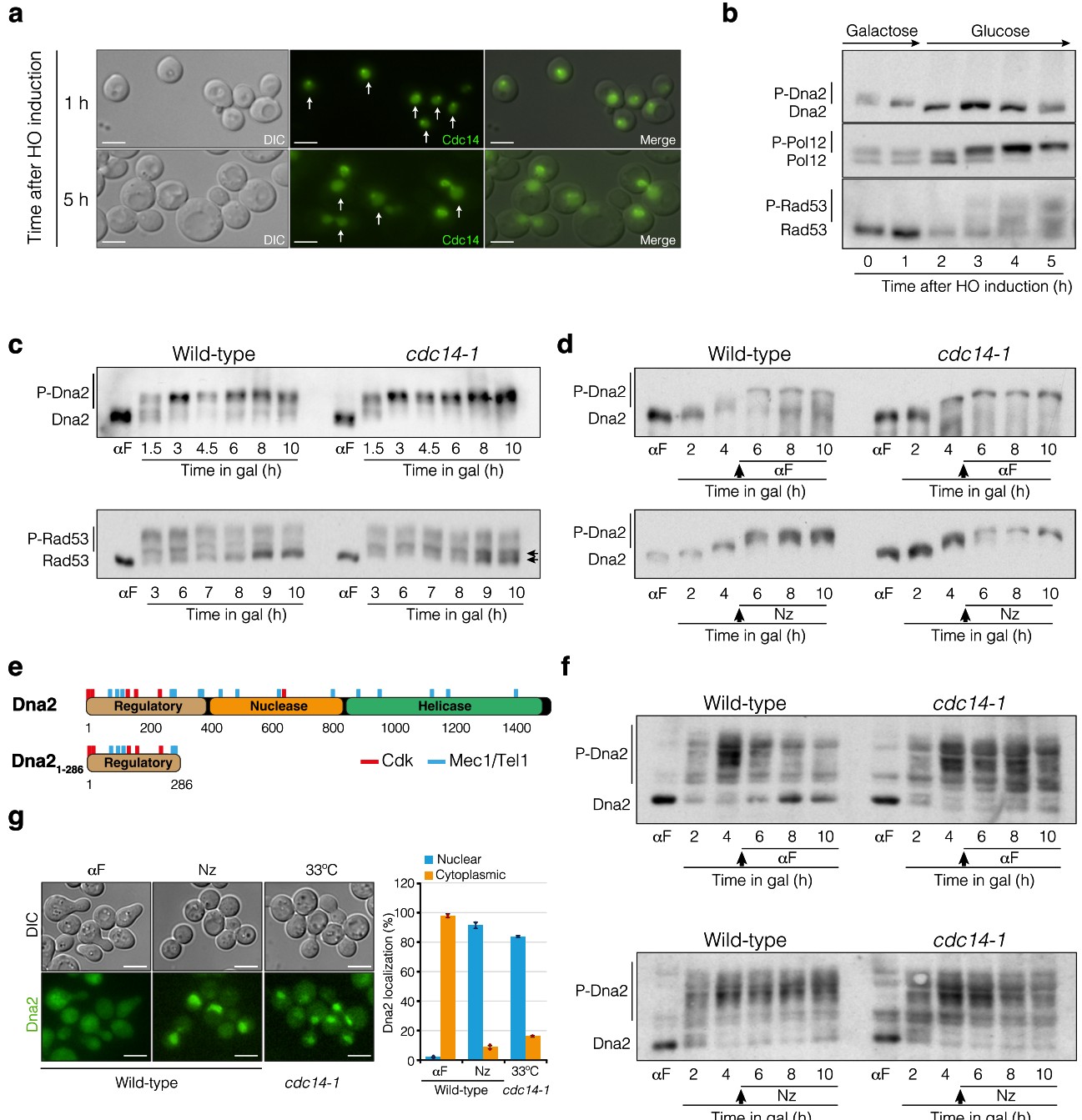

**Fig. 7 | Mitotic Cdc14 phosphatase dephosphorylates Dna2. a** Over-expression of Cdc14 leads to its accumulation in the nucleoplasm. After overnight culture of *cdc14-1* PMV cells in YP-Raffinose at 25 °C, HO was expressed together with a *GAL1-GFP-CDC14* version, and the culture was transferred to 33 °C to inactivate the endogenous *cdc14-1* allele. After 1.5 h, glucose was added to repress *CDC14* expression and pictures were taken at discrete intervals to follow Cdc14 localization. Representative pictures at 1 h and 5 h after HO induction are shown. Arrows indicate cells with Cdc14 accumulated at the nucleoplasm. Scale bar: 3 µm. **b** Dna2 or Pol12 were tagged with HA in a *cdc14-1 GAL1-GFP-CDC14* PMV strain to follow their phosphorylation levels by western blot under the same experimental conditions used in **a**. An anti-Rad53 antibody was used to determine Rad53 phosphorylation levels along the experiment. **c** Dna2 phosphorylation dynamics in wild-type and *cdc14-1* PMV cells after the induction of an HO-break. After overnight culture of cells containing the endogenous *DNA2* tagged with HA in YP-Raffinose, cells were blocked in G1 by using the pheromone alpha factor (αF) and released in the presence of galactose to induce HO expression. Samples were taken at the indicated time points to follow Dna2 (top panel) and Rad53 (bottom panel) phosphorylation by western blotting. A concentration of 7 µM of Phos-Tag was used to facilitate the

separation of the Dna2 phosphobands. Black arrows denote a differential phosphorylation state of Rad53 in *cdc14-1* cells **d** Overnight YP-Raffinose cultures of wild-type and *cdc14-1* PMV cells were processed as in **c**, but alpha factor (top panel) or nocodazole (bottom panel) was added after 5.5 h after HO induction. Samples were taken at the indicated time points to follow Dna2 phosphorylation. A concentration of 7 µM of Phos-Tag was used to facilitate the separation of the Dna2 phospho-bands. **e** Schematic representation of full length and a truncated Dna2₁-₂₈₆ version illustrating the Cdk and Mec1/Tel1 consensus sites. An amino acid residue number scale is shown. **f** Overnight YP-Raffinose cultures of wild-type and *cdc14-1* PMV cells carrying the truncated version Dna2₁-₂₈₆ were processed as in **d** but in the absence of Phos-Tag. Samples were taken at the indicated time-points to determine Dna2 phosphorylation. **g** Overnight YP-Raffinose culture of wild-type PMV cells carrying the endogenous *DNA2* tagged with GFP was treated with alpha factor (αF) or nocodazole (Nz) to block cells in G1 or metaphase, respectively. The same strain but in a *cdc14-1* background was used to check for Dna2 localization after switching to restrictive temperature. Scale bar: 3 µm. Graph on the right represents the percentage mean ± SD from two biologically independent experiments of cells with nuclear (blue) or cytoplasmic (orange) Dna2.

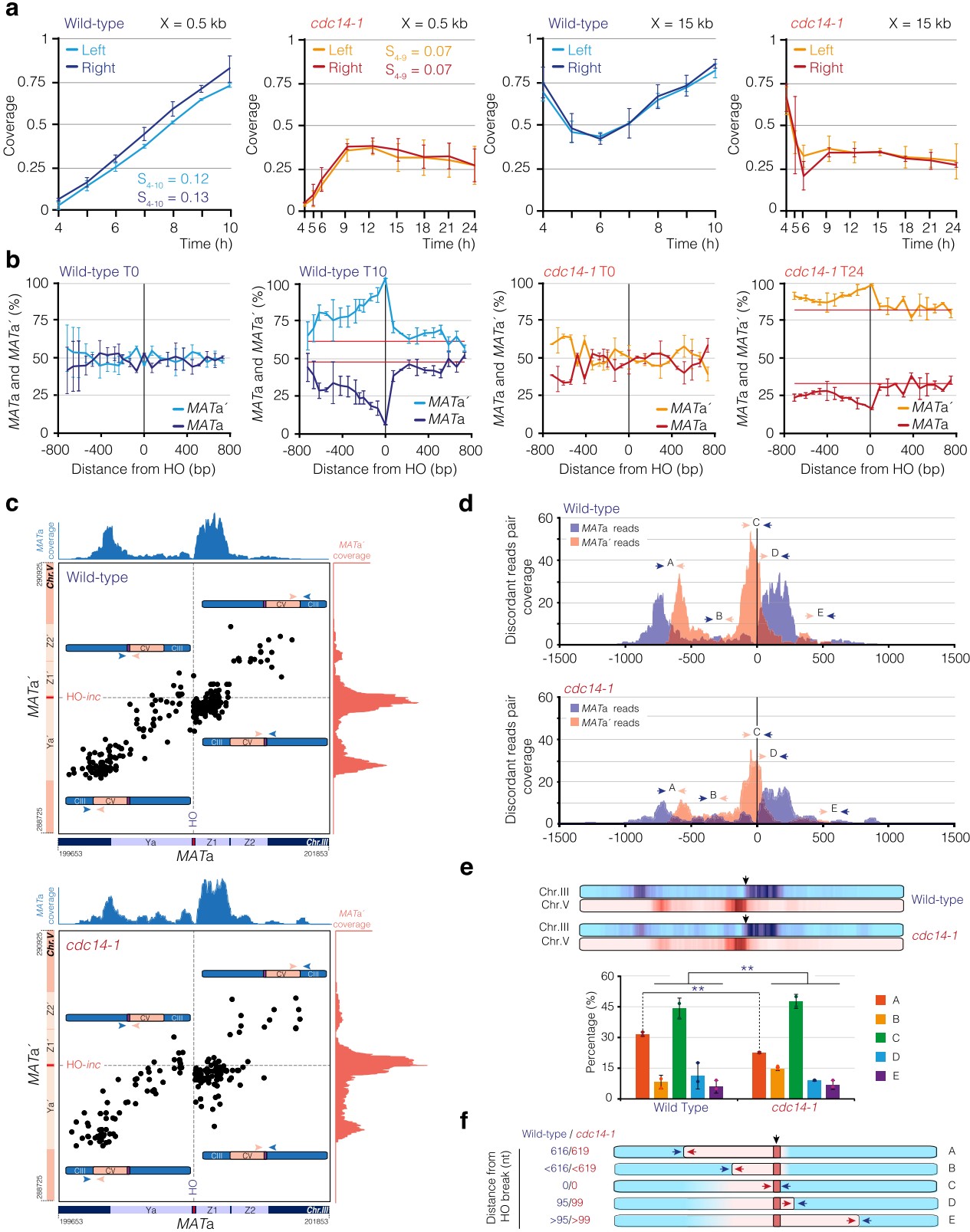

## FACS

DNA content was determined by staining cells with propidium iodide and analyzing them in a BD FACSCalibur™ flow cytometer (BD Biosciences). Data were processed by using a CellQuest™ Pro v6.0 software. A minimum of 10.000 cells were scored per time-point.

## Chemiluminescent Southern blotting

10 ml of cell cultures at an $OD_{600}$ of 0.4 were harvested by centrifugation and washed with 1 ml PBS. After centrifugation, pellets were flash frozen and stored at −80 °C. Cell lysis was performed by treating the pellets with 40 units of lyticase in DNA preparation buffer (1% SDS, 100 mM NaCl, 50 mM Tris-HCl, 10 mM EDTA) for 10 min. DNA was

**Fig. 8 | Cdc14 facilitates DNA re-synthesis and extension of GC tracts.**
**a** Coverage levels at 0.5 kb (left panels) and 15 kb (right panels) at both sides of the HO cleavage site in wild-type (blue) and *cdc14-1* (red) PMV cells from 4 h after the endonuclease expression. Synthesis rates (S) between different time intervals are indicated (see Supp. Figure 6b for details). Light blue/red: left HO side. Dark blue/red: right HO side. The graphs show the mean ± SD from two biologically independent experiments. **b** Relative proportion of reads containing polymorphisms in the *MAT*a and *MAT*a' regions of chromosomes III and V, before (T0) and after (T10/T24) induction of the HO endonuclease in wild-type (blue) and *cdc14-1* cells (red). Horizontal red lines mark the expected profile of polymorphisms in *MAT*a and *MAT*a' considering the total DNA repair levels obtained by 10 h (wild-type) and 24 h (*cdc14-1*). The graphs show the mean ± SD from two biologically independent experiments. **c** Graphs representing the coordinates between *MAT*a and *MAT*a' discordant read pairs in wild-type (top) and *cdc14-1* cells (bottom). Intersection of *MAT*a/*MAT*a' coordinates on each quadrant represent distinctive GC boundaries (see diagrams). Discordant reads pairs falling at the left-bottom quadrant and right-bottom quadrant correspond, respectively, to the left and right boundaries of GC events taking place at the left side of the HO break. Discordant reads pairs falling at the left-top quadrant and right-top quadrant correspond, respectively, to the left and right boundaries of GC events taking place at the right side of the HO break.

The top blue and right red coverage profiles are generated by filtering *MAT*a and *MAT*a' reads from the discordant pairs and aligning them to the reference genome. **d** Overlap of the *MAT*a and *MAT*a' coverage profiles obtained from the analysis shown in **c** in wild-type (top panel) and *cdc14-1* (bottom panel) cells. Blue and red represent, respectively, the read coverage profile of *MAT*a and *MAT*a' reads isolated from the *MAT*a/*MAT*a' discordant read pairs identified. A, B, C, D, and E represent the relative position of the discordant read pairs along the *MAT locus* after repair. **e** Blue and red densitometry panels displaying coverage levels of both *MAT*a and *MAT*a' *loci*, respectively. Graph represents the percentage of A-E discordant read pairs relative to the total number of events. Graph shows the mean ± SD from two biologically independent experiments. *P*-values were calculated using a two-tailed unpaired Student's *t*-test. *: $P \leq 0.05$, **: $P \leq 0.01$; ***: $P \leq 0.001$. Black arrows mark the position of the HO cleavage sites. **f** Measurement of the maximum extension of GC at both sides of the HO break in wild-type and *cdc14-1* PMV cells. The coverage profiles shown in **d** were used to determine the boundaries of GC generated during the incorporation of *MAT*a' polymorphisms into the *MAT*a *locus* in wild-type (blue) and *cdc14-1* (red) cells. A, B, C, D, and E represent the relative position from the HO site (nucleotides) of the discordant read pairs along the *MAT locus* after repair. Black arrow marks the position of the HO cleavage sites.

extracted by incubating with phenol:chloroform:isoamylalcohol (25:24:1) for 10 min, and after centrifugation, the aqueous fraction was precipitated with ethanol and resuspended in TE buffer. Genomic DNA was digested with *Eco*RI, *Sty*I or *Kpn*I separated on 1% agarose gels and subjected to Southern blotting. Probes were generated by labeling a PCR-amplified DNA fragment with a mix of nucleotides containing fluorescein-12-dUTP (Fluorescein-High Prime, Roche, 11585622910). The oligonucleotides used to synthesize the probes are listed in Supp. Table 2. Detection was achieved by using an anti-fluoresceine antibody conjugated to alkaline phosphatase (Anti-Fluorescein-AP Fab fragments, Roche, 11426338910) at a 1:250,000 dilution. Membranes were incubated with CDP-Star detection reagent (Amersham, RPN3682) and exposed to films. The processing and analysis of the images were performed by using the FIJI software (https://imagej.net/Fiji).

### Western blots
Samples for western blotting were prepared by trichloroacetic acid (TCA) extraction. Cells were collected, centrifugated, and fixed with 20% TCA. Cell lysates were obtained by breaking with glass beads in a FastPrep homogenizer (MPBio) (three 20 s cycles at power setting 5.5). Proteins were precipitated by centrifugation at 2,200 g for 5 min at 4 °C, and the pellet was solubilized in 1 M Tris-HCl (pH8) and SDS-PAGE loading buffer. After boiling for 10 min, the insoluble material was removed by centrifugation and the supernatant was loaded onto 6% acrylamide gels. To facilitate the separation of the Dna2 phospho-bands, a final concentration of 7 µM of Phos-Tag (Wako, 300-93523) was used. Proteins were transferred onto a PVDF membrane (Hybond-P, GE Healthcare, 15259894) and blocked with 5% milk in PBS-Tween (0.1%). Anti-Rad53 antibody was used at a 1:1,000 dilution (Abcam, ab104232) and the secondary anti-rabbit antibody was used at a concentration of 1:5,000 (GE Healthcare, NA934). Anti-HA antibody was used at a 1:2,500 dilution (Merk, 11666606001). Anti-MYC antibody was used at a 1:2,500 dilution (Merk, C3956). Secondary anti-mouse antibody was used at a concentration of 1:25,000 (GE Healthcare, NA931). After washing the membranes in PBS-Tween, they were incubated with SuperSignal® West Femto (Thermo Scientific, 10391544), followed by exposure to ECL Hyperfilm (GE Healthcare, 28-9068-37).

### Real-time PCR and D-loop analysis
Real-time PCR to determine the stability of the ssDNA generated after resection was performed by using two primers located at -1.5 kb from the HO break (-1.5 kb HO Fwd, GTCTGTGATGTTGGAGATATG and -1.5 kb HO Rev, AGAATTAGCGGACCTCTTGAG). The dynamics of D-loop extension was accomplished by PCR and qPCR amplification

using a primer distal to the *MAT*a *locus* (Mata Distal 2, AGGATGC CCTTGTTTTGTTTACTG) and a primer within the Ya' at the *MAT*a' *locus* (Arg_Ya´HO_Arg check Fwd, TACCAGCCAACATCAGTGTAG). Amplification with a pair of primers situated in the *ACT1 locus* was used for DNA content normalization (Act1 for KpnI cut Fwd, CAGGTATTGCC GAAAGAATGC; Act1 for KpnI cut Rev, GTCCCTGAGATGAGTAAGATC).

### Genomic sequencing and data analysis
Genomic libraries were generated by using the Nextera XT DNA Library Preparation Kit (Illumina). Briefly, 10 ng of DNA obtained from the DNA extraction (see above) were incubated at 55 °C for 5 minutes in an enzymatic tagmentation reaction to fragment DNA and add adapter sequences. A 7-cycle PCR program was used to amplify the tagmented DNA and to add indexes and sequences required for cluster formation. DNA libraries were purified using Sera-Mag® magnetic beads (Cytiva, 29343052) and analyzed on an Agilent 2100 Bioanalyzer using a High Sensitivity chip. The concentration for each library was determined by measuring in a Qubit 3.0 fluorometer. 2.1 pM of a genomic DNA pool containing all the libraries and a 1.8% of PhiX library as a control was used to follow a paired-end 75 bp sequencing protocol in an Illumina NextSeq500 platform. Sequencing coverage in different experiments ranged from 26 to 88 per nucleotide after quality filtering. Sequenced reads were aligned to the *S. cerevisiae* PMV reference genome (Supplementary Data 1, PMV_RG.fasta) developed in[40] with Bowtie (v1.0.0)[74] using the -v 0 alignment mode (allowing 0 mismatches), and the -k 1 reporting mode (allowing a maximum of 1 valid alignments per read). The retrieved sam files were converted to bam files by using the Samtools (v1.12) utility for sequence alignment[75]. Alignment bam data were normalized using reads per genomic content (RPGC) by using the Deeptools (v3.5.1) utility. The generated BedGraph files were used to calculate the coverage along the chromosomes in the PMV background strain (Bedtools, v2.30.0). For visualization of the BedGraph coverage profiles we used the Integrative Genomics Browser IGB (v9.0.2)[76].

Normalized read coverage profiles were obtained in a sequential two-step approach. First, a threshold of 0.2 was set up in the T0 sample to eliminate from the analysis those genomic regions with low intrinsic coverage. These low-coverage regions at T0 were also eliminated from the analysis of the whole set of samples collected after HO expression. Second, each time point was normalized against the 0 h sample. The normalized read coverage profile obtained was averaged in 100 nt bins and plotted.

For the normalization of 2D/3D read coverage profiles and colormaps, the averaged coverage of 2 kb sections along the

chromosomes from samples at different time points after HO induction was normalized against the same sections of the 0 h sample and to a 120 kb region between coordinates 210000-330000 of chromosome V by applying the formula $C_N = [(CIII_{Tx} / CIII_{T0}) / (CV_{Tx} / CV_{T0})]$, where $C_N$ is the normalized coverage, $CIII_{Tx}$ and $CIII_{T0}$ are the averaged coverage of each 2 kb sections from chromosome III at $T_x$ or $T_0$ timepoints, respectively, and $CV_{Tx}$ and $CV_{T0}$ are the averaged coverage of the 120 kb section from chromosome V at $T_x$ or $T_0$ timepoints, respectively. While T0 normalization corrects the data relative to the undamaged sample, chromosome V normalization allows comparison between different experiments/strains. Colormaps were generated by translating read coverage values from the 3D normalized read coverage profiles into a set of colors ranging from red (high coverage) to purple (low coverage).

Short-range resection was displayed as the normalized read coverage profile of a $\pm 100$ nt region from the HO cleavage site 1 h after the HO induction. Long-range resection at both sides of the HO-induced break was estimated by calculating the correlation between the distance from the HO cleavage site where read coverage dropped to 0.5 and the time needed to reach that particular value from 1 h to 4 h after inducing the HO. The resection rate (R) was determined by applying the formula $v = \Delta s / \Delta t$ in the 1-4 h interval.

Maximum extent of resection was calculated by measuring the distance from the HO-induced break at which the read coverage reached a cut-off value of 0.9 relative to the whole chromosome III coverage average in the normalized 2D coverage profiles for each time point after the HO induction. Note that, since the average coverage of chromosome III diminishes with resection, the cut-off values for those samples enriched in ssDNA tend to decrease.

Synthesis rate (S) was calculated by using the trend line equations ($y = mx + b$) from the normalized read coverage profiles with a fixed $x$-value of 500 and 15,000 (0.5 kb and 15 kb, respectively) to calculate the $y$-value coverage.

DNA re-synthesis initiation rate was obtained from the formula $v = \Delta s / \Delta t$, where $\Delta s$ was set to 0.5 (distance from the HO cleavage site) and $\Delta t$ was calculated by using the trend line equations obtained from the synthesis coverage data (Wt: 4 h–10 h; cdc14-1: 4 h–9 h), to determine the timing needed to progress from 0 to a 0.5 coverage value.

For gene conversion analysis, paired-end sequence reads were aligned individually as single reads using Bowtie to the *S. cerevisiae* PMV reference genome using the -v 0 alignment mode (allowing no mismatches), and the -k 1 reporting mode (allowing only 1 valid alignment per read). Coverage for every polymorphic coordinate across each *MAT loci* was calculated at 0 h and 10 h in the wild-type or 0 h and 24 h in the *cdc14-1* mutant. A correction factor was needed since DNA repair was not 100% efficient in the 10 h and 24 h samples. The values of these correction factors correspond to the maximum levels of coverage recovery measured at 0.5 kb distance from the HO in wild-type and *cdc14-1* cells.

Discordant read analysis was performed by aligning reads with Bowtie 2 (version 2.4.4)[77] using the--score -min C,0,0 -N 0 and--end-to-end read alignment (allowing 0 mismatches). Discordant read pairs were filtered by selecting those in which reads from each pair aligned at different chromosomes. Reads of 75 nt length were filtered, and multi-mapped reads were removed. Retrieved sam files were converted to bam files by using the Samtools (v1.12) utility, normalized using RPGC (Deeptools, v3.5.1) and the obtained BedGraph files were used to calculate the coverage along chromosomes III and V in the PMV background strain (Bedtools, v2.30.0).

## Microscopy
In vivo fluorescence microscopy of GFP and 4′-6-diamidino-2-phenylindole (DAPI) staining was performed with a Deltavision microscope (PersonalDV; Imsol) equipped with a Cool Snap HQ CCD camera (Photometrics). A maximum projection image obtained by merging five z–planes pictures was used to determine the average number of Dna2 foci. At least 100 cells were scored per sample. Images were visualized and analyzed by using the FIJI software (v1.0).

## Statistics and reproducibility
All statistical analyses were performed using GraphPad Prism. *P*-values were calculated using a two-tailed unpaired Student's *t*-test. Significance was set at $P \le 0.05$ for all experiments. The number of replicates, statistical tests applied and *P*-values for each analysis are included in the figure legends. All Southern blots and micrographs images shown in the article were performed in a minimum of two biological replicates to ensure the reproducibility of the results.

## Reporting summary
Further information on research design is available in the Nature Portfolio Reporting Summary linked to this article.

## Data availability
Genomic datasets for the wild-type strain were generated by[40] and deposited in the Sequence Read Archive (SRA) database under accession code PRJNA785778. Genomic datasets for the *cdc14-1* mutant have been deposited in the SRA database under accession code PRJNA877059. Source data are provided in this paper.

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

## Acknowledgements

We thank Francisco Antequera for the generation of the genome-wide datasets. We also thank Pedro San Segundo, Mónica Segurado, Cristina Martín, Alfonso Fernández, and Jesús Carballo for discussions and valuable comments on the manuscript. We also thank the members of our laboratories for their helpful discussions. This work was supported by the projects BFU2016-77081-P, PGC2018-097963-B-I00, and PID2021-125290NB-I00, funded by the MCIN/AEI/10.13039/501100011033/ and by the "FEDER, Una manera de hacer Europa", awarded to A. C-B. The IBFG is supported in part by the institutional grant from the Goverment of Castile and Leon "Programa "Escalera de Excelencia" de la Junta de Castilla y León, Ref. CLU-2017-03 co-funded by P.O. FEDER de Castilla y León 14-20" and by the Internationalization Project "CL-EI-2021-08-IBFG Unit of Excellence" of the Spanish National Research Council (CSIC), funded by the Regional Government of Castile and Leon and co-financed by the European Regional Development Fund (ERDF "Europe drives our growth"). A.C. was the recipient of a predoctoral fellowship from the "Junta de Castilla y León". F.R. was the recipient of a predoctoral fellowship from the "Ministerio de Economía y Competitividad". L.I. was the recipient of a "Jae-INTRO" grant from the CSIC and a predoctoral fellowship from the "Junta de Castilla y León". C.D. was the recipient of a predoctoral fellowship from the "Universidad de Salamanca". A.E.-M. was the recipient of a predoctoral fellowship from the "Junta de Extremadura" financed by the European Social Fund (ESF) PD18066.

## Author contributions

A.C., F.R., L.I., C.D., E.M., A.E.-M., and J.C.-B. performed all the experiments. L.I. collaborated in the analysis of the sequencing data. A.C.-B. designed the research plan, supervised the experiments, collaborated in the analysis of the data, and wrote the manuscript. All authors discussed the results and approved the final version of the manuscript.

## Competing interests

The authors declare no competing interests.
