## [Peer Review File · Nature Communications]

Cdc14 phosphatase counteracts Cdk-dependent Dna2 phosphorylation to inhibit resection during recombinational DNA repairREVIEWER COMMENTS

Reviewer #1 (Remarks to the Author):

Overview

The manuscript details the analysis that loss of the Cdc14 phosphatase has on the rates/extent of DNA end resection and recombination-dependent DNA resynthesis in *S. cerevisiae*. The experiments employ an inducible site-specific DNA break created by the HO endonuclease, and perform analyses in a number of genetic situations including abrogation of one of the key resection proteins, Dna2. The study employs a combination of site-specific Southern blotting and whole-genome copy number sequencing to infer resection and resynthesis rates. The authors use their observations to support the view that a role of Cdc14 is to restrict hyper-activity of the Dna2 nuclease (activated by phosphorylation).

Overall, the data are extensive and extremely detailed. However, in many places the presentation of data and its description within the text could be made clearer. A simple change in the main text would be to break down some of the large paragraphs within sections into 2-3 times as many shorter paragraphs, where each is focused on describing a particular hypothesis/experiment. This will permit the logical thread to be much more easily followed.

In addition to this minor point (which is easily addressable), there are some technical and interpretative concerns—namely:

1. MINOR: Much of the analysis pivots upon effects arising many hours (6+ hours) after a DNA break has been generated. Is this timepoint representative of a physiological stage that would happen in wild type cells in any natural situation? At least discussing/commenting on this aspect would be of help to the readers.

2. MAJOR: Although a direct connection of Cdc14 to resection control (via Dna2) is suggested, it seems to me that the authors cannot rule out the equally valid and indeed realistic possibility, that the hyper-resection observed when Cdc14 is inhibited is a consequence of a DNA repair delay, and that Cdc14 phosphatase is actually involved in promoting repair (and only involved indirectly in resection). i.e. The cause and effect are the reverse of what is proposed. At the very least this concept needs either discussing or refuting (and if refuted, with evidence to the contrary).

Note: I am not arguing that Dna2 is not involved in the hyper-resection—the data are clear—I am suggesting, instead, that the role that Dna2 is playing is a consequence of an independent defect caused by Cdc14 loss.

General:

It would help if the authors were clear in the abstract what species/system the experiments were performed in (Here we show, in *S. cerevisiae*, that the lack...)

It would also help if the introduction were clear as to which aspects are generalised (across species) and which are unique to the molecular details elucidated from yeast studies. For example is Cdc14 a conserved phosphatase with conserved roles?

Specific technical queries:

1. MINOR: Why were some experiments performed in G1-arrested cells (early sections) and some in exponential cultures? (Later sections.) Is there an expectation of a difference? Please clarify.

2. MINOR: What does it mean when the genome coverage data was normalised both against t=0 and 120 kb region on chromosome V? Please explain the logic/rationale here. And what does "normalised" mean in those context? Division? Subtraction?

3. MINOR: Fig 1 BC. It would help readability greatly if the small horizontal cartoons/diagrams with the blue circles/red bars in them were labelled with text that describes what they are. e.g. Uncut parent, gene conversion (GC) products, etc.

Secondly, the juxtaposition/interleaving of the low and high exposure images—especially when the

cropped areas are different makes it very challenging to understand the image. I think that the long exposure alone (without any cropping of the gel) would make the raw data a lot easier to understand.

4. MINOR: Fig 2. What explanation do the authors have for why the coverage for the most proximal probes returns to a higher level before that of the probes furthest away from the DSB? What repair mechanism may account for this?

5. MINOR: Fig 3B. The presentation style here is unhelpful and distracting. Whilst a heatmap might be useful, the use of the faux/angular peaks/shading has no merit or utility. A simple heatmap of coverage through time would be much easier to visually understand.

6. MINOR: Fig 3C. Line 533. Has it been demonstrated that this assay for coverage ± 100 nt flanking the HO cut site is indeed a measure of short-range resection? It seems to this reviewer that cleavage alone (without any resection) might reduce the probability of generating coverage in this region because the genome would be broken, and thus prevent any molecules/reads from spanning the junction. To understand this, has the assay been performed in conditions in which resection is completely inhibited (exo1D, dna2 mutant and sae2D?). If so, what were the results?

Note that, at most, resection here should presumably only reduce the coverage to 50% of maximum (based on loss of one strand), whereas graphs appear to show coverage dropping below 0.5 and close to zero in proximity of the DSB.

7. MINOR: Fig 3E. Line 542. How was the "maximum extent of resection" calculated/defined?

8. MAJOR: Fig 6A Lines 240-246. Any differences in wild type \pm nocodazole or alpha factor were very hard for this reviewer to see. Secondly, given that Cdc14 inhibition has the same effect in both arrests, I am unclear how the authors then conclude that Cdc14 mainly inhibits resection during its mitotic activation. This part needs substantial revision/clarification to make sense.

9. MINOR: Fig 6B. Lines 352. Please clarify how the lack of discordant pairs between II and VIII validates the assay? What is special about the III-VIII junction? Can a diagram of this locus/junction be provided to explain this control?

10. MAJOR: Fig 8C is unintelligible. I am sure there are interesting data here, but the figure is too poorly explained to be understood by this reviewer. What purpose do all the inset cartoon chromosome bars serve/indicate?

11. MAJOR: Line 364. Please explain how the directionality (and especially extent) of GC can be inferred from the coordinates. This sounds interesting/exciting, but the lack of description here prevents any critical understanding/appraisal. How long were the reads? Was there allele information across the entire fragment, or are their (expected) gaps on individual molecules between the (short) paired reads? Does this latter effect affect the data/analysis?

12. MAJOR: Fig 8E. Lines 370-386. I am not convinced there are any differences \pm Cdc14 here. No statistics are provided. Moreover, even if they were, there appears to be no hypothesis being tested. The fact that some regions (may) show differences, and others not, may just be experimental noise. Without a hypothesis, multiple-test-corrected statistical thresholds need to be employed.

Reviewer #2 (Remarks to the Author):

This manuscript by Campos et al. presents a new control mechanism of Dna2-mediated resection by Cdc14 phosphatase. It is shown that Cdc14 phosphatase is needed for efficient DSB repair. Interestingly initial and extensive resection, strand invasion, and even initial DNA synthesis appear to be fine, but cells fail to complete DNA synthesis in Cdc14 deficient cells. This problem is

suppressed at least partially by the elimination of Dna2 or by making a Dna2-Cdk1 phosphorylation mutant. Thus it seems that less Dna2 activity is helpful toward the end of DSB repair, and this is accomplished by Cdc14 phosphatase. In general, it is interesting that the authors follow resection after strand invasion, which so far was very little studied. The idea that Dna2 activity has to be decreased during repair is novel and interesting, but the work needs significant improvements.

Questions to be addressed:

Cdk1 controls the localization of Dna2. In cells that lack Cdc14 phosphatase activity, Dna2 localization should remain in the nucleus throughout the cell cycle. A clear analysis of Dna2 localization without any DSB is needed for control by Cdc14.

The new role of Cdc14 phosphatase is proposed in cells arrested in the G2/M phase in response to DNA damage. Cdc14 is known for its role in mitotic exit/anaphase. Is it known that Cdc14 is active earlier in G2/M arrested cells upon damage response?

Please provide more information on the *cdc14-1* allele; how do cells arrest or respond in general to increased temperature but without a DSB? Do they arrest and replicate normally? In Figure S1, it would be good to indicate when the temperature is raised to 33°C. Would cells stay arrested in *cdc14-1* at high temperatures? How this affects the conclusions?

Line 122 and further

D-loop intermediate measurement – this assay measures initial DNA synthesis by PCR, as PCR product is detected only after initial DNA synthesis. It is interesting that initial DNA synthesis is completed, but further synthesis is not possible. This should be discussed. Also, the analysis should be done rather by qPCR to see the possible difference in *cdc14-1* cells at the nonpermissive temperature.

The lack of repair is not due to excessive resection in *cdc13-1* as resection is comparable in WT and *cdc14-1* cells – Figure S1D. Here resection was measured in the absence of DSB repair, and it seems to be the same or even slower in *cdc14-1*. It would be good to plot WT and *cdc14-1* on one plot to see the resection slowness of *cdc14-1*. Line 166 should say, “over-resect the two sides of repairable HO break”. In nonrepairable break, there is no over-resection in *cdc14-1*.

Dna2 depletion in DNA2-AID *cdc14-1* stain rescues the repair defect. Would SGS1 deletion (*Sgs1* works with Dna2 in resection) suppress *cdc14-1* in a similar way as DNA2-AID? It is an important experiment to validate the model.

The authors suggest that excessive resection is toxic in Cdc14 minus cells, but it could be that more resection is observed because there is no repair, and less resection can bypass the toxicity of Cdc14 depletion (whatever the reason is). To test this possibility, it would be good to limit resection in a different way, such as *fun30* deletion.

Line 208. Dna2 is not activated in response to Cdk-dependent phosphorylation. It is simply phosphorylated by Cdk1 in a cell cycle-dependent manner. This phosphorylation regulates its nuclear localization and in consequence, recruitment to DSBs.

Minor:

The topic of the manuscript is complex and needs attention to describe experiments in a simpler, easy-to-follow way.

In the introduction line 40, please add information on the role of Dna2 phosphorylation in the regulation of nuclear localization (Kosugi et al., 2009).

Fig 1AB is hard to follow. Perhaps adding the sizes of all possible products in the diagram and indicating which one is which in B would be helpful.

Line 101 – indicate “initial” resection, as the disappearance of the cut band only informs about initial resection.

Lane 110 – please modify the second part of the phrase. Do you mean “later stage of repair” or “completion of repair”? As stated, it is not clear.

Resection during repair in a strain nearly identical to the one studied here was analyzed in a similar way by Chung et al. 2010 (PMID: 20485519). These results should be discussed.

The rationale for an experiment in 6A should be better explained for people who do not study Cdc14.

Reviewer #3 (Remarks to the Author):

In the manuscript "Cdc14 phosphatase counteracts Cdk-dependent Dna2 phosphorylation to inhibit resection during recombinational DNA repair" Andres Clemente-Blanco and colleagues investigate the role of the mitotic phosphatase CDC14 in DSB repair by homologous recombination (HR) using the *Saccharomyces cerevisiae* model. Studying the repair of an HO-induced break at the MAT locus using an elegant system where the recombination template is distinguished by 23 polymorphisms they find that CDC4 is a regulator of DNA end resection. Specifically, CDC14 inactivity leads to hyper-resection that the authors link to hyperactivity and phosphorylation of the DNA2 nuclease. Notably, this hyper-resection is suggested to lead to a block of repair by HR downstream due to a block to DNA re-synthesis. This paper is intriguing for two reasons – first, it indicates for the first time that CDC14 is an antagonist of DNA end resection and cell-cycle regulated phosphorylation of DNA2. Second, it suggests that hyper-resection is inhibitory to DNA repair. As such, the paper is clearly interesting for the broader field, but several technical points need to be addressed prior to publication.

Major points:

1 – Reproducibility. For the majority of the experiments shown in the paper, it was not entirely clear if they were replicated. In fact, lack of statements on biological replication suggests they are not. While I can see that in a major study such as this not all experiments can be replicated, this needs to be done at least for key experiments. This is especially true, since not all effects are as “black-and-white” as the experimental description suggests.

2 – The effect of the CDC14 mutant is intriguing, but they are only shown for a single system at a single genomic location. The text of the paper does however suggest that these findings can be generalized. At least the hyper-resection phenotype should be reproduced at another genomic location or alternatively the text would need to be strongly adapted to this shortcoming.

3 – The resection assays in Fig. 2, 4, 5 and 6 are not easy to interpret for the non-expert as there is a biphasic response of resection and re-synthesis. As a strength of the system is that repair can be unambiguously quantified, I suggest to plot repair kinetics next to these graphs. Also, I miss plots of resection vs distance from DSB.

4 – The quantifications are often poorly labelled, particularly in Fig. 3. While I appreciate the use of cartoons as labelling, an additional labelling of the plots is needed. Additionally, figure legends need to be rewritten. They have sufficiently detail about experimental procedures, but they do not at all clarify what is being shown where. The combination of these two factors makes it often difficult to understand what is actually shown in which figure.

5 – Figure 4 and 5 show a lot of pairwise comparisons in the individual figure panels and the authors do not clarify whether additional comparisons are possible. However, additional comparisons would be very useful, for example CDC14-1 vs CDC14-1 DNA2 add back (Fig. 5) or

CDC14-1 DNA2-AID vs CDC14-1 DNA2-phosphomimetic (Fig. 5).

Minor points:

6 – Fig.8 I find the comparison of different time points problematic, T10 h for WT and T24 h CDC14-1. It is also not clear to me why those timepoints were chosen. According to Fig. 8A not much DNA synthesis happens in CDC14-1 after 10 h.

7 – Fig. 6A There does not seem to be much of a difference between the nocodazole and alpha-factor treatment of WT cells. This seems at odds with the authors conclusion.

8 – Fig. 4C There seems to be only a partial defect between EtOH and IAA treatment. Can the authors please comment on this?

9 – Fig. A-B I am not sure whether a comparison is possible, but if it is, it seems like the EXO1 mutant appears to have enhanced resection, which would be different from other findings.

10 – Fig. 1B-C In the graphs blue labelling of graph and legend appears not to match.

Response to Reviewer #1:

1) *“Overall, the data are extensive and extremely detailed. However, in many places the presentation of data and its description within the text could be made clearer. A simple change in the main text would be to break down some of the large paragraphs within sections into 2-3 times as many shorter paragraphs, where each is focused on a describing a particular hypothesis/experiment. This will permit the logical thread to be much more easily followed.”*

As suggested by the reviewer, we have shortened the paragraphs to facilitate the comprehension of the experiments shown in the manuscript.

Technical and interpretative concerns

1) *“MINOR: Much of the analysis pivots upon effects arising many hours (6+ hours) after a DNA break has been generated. Is this timepoint representative of a physiological stage that would happen in wild type cells in any natural situation? At least discussing/commenting on this aspect would be of help to the readers.”*

I agree that ectopic recombination in the PMV is a rather slow process compared with other allelic recombination system. However, the same kinetic has been reported in other DNA repair systems extensively used for the analysis of inter-chromosome recombination, as the tGI background (see for example Ira G., et al., Cell, 2003). In fact, we engineered the PMV strain (Ramos F., et al. Cell Reports, 2022) by mimicking the genomic positions of the main tGI features (i.e., *MAT* locus on Chr.III and the *ARG5,6* locus on Chr. V as recipient/donor chromosomes for HR) to resemble its DNA repair dynamics, thus making both strains comparable between them.

Besides, ectopic recombination repair kinetics relies on multiple factors, such as the length of the donor sequence, the proximity between the recipient and the donor chromosome and the changes in chromatin structure and chromosome movements (please, see Haber, JE, BioEssays, 2018). All this could affect the dynamics in inter-chromosomal DNA repair. We have commented on these notions in the discussion section of the manuscript to clarify that our DNA repair kinetics might be different from other DNA repair systems (lines 490-496).

2) *“MAJOR: Although a direct connection of Cdc14 to resection control (via Dna2) is suggested, it seems to me that the authors cannot rule out the equally valid and indeed realistic possibility, that the hyper-resection observed when Cdc14 is inhibited is a consequence of a DNA repair delay, and that Cdc14 phosphatase is actually involved in promoting repair (and only involved indirectly in resection). i.e. The cause and effect are the reverse of what is proposed. At the very least this concept needs either discussing or refuting (and if refuted, with evidence to the contrary). Note: I am not arguing that Dna2 is not involved in the hyper-resection—the data are clear—I am suggesting, instead, that the role that Dna2 is playing is a consequence of an independent defect caused by Cdc14 loss.”*

I absolutely agree with the reviewer that, although it seems clear that Cdc14 negatively controls resection progression by targeting Dna2, we cannot discard other functions for the phosphatase in the repair pathway. Accordingly, degradation of Dna2 does not fully restore the over-resection phenotype of *cdc14-1* mutant cells. This result, together with the fact that multiple components of the resection machinery are regulated by Cdk, suggests that Cdc14 could have multiple roles in DSB repair that might directly or indirectly affect resection dynamics. As recommended, we have included two paragraphs in the discussion commenting on these possibilities (lines 505-508 and 517-524).

General comments

*“It would help if the authors were clear in the abstract what species/system the experiments were performed in (Here we show, in *S. cerevisiae*, that the lack...)”*

We have now specified in the abstract (line 3) and in the introduction (line 47 and 64) that this work has been performed using *S. cerevisiae* as model organism.

“It would also help if the introduction were clear as to which aspects are generalised (across species) and which are unique to the molecular details elucidated from yeast studies. For example is Cdc14 a conserved phosphatase with conserved roles?”

We have now remodeled the introduction to specify the particular model organisms in which the different roles for Cdc14 have been identified (see for example line 48, 50-51, 58 and 61).

Specific technical queries

1) *“MINOR: Why were some experiments performed in G1-arrested cells (early sections) and some in exponential cultures? (Later sections.) Is there an expectation of a difference? Please clarify.”*

I agree with the reviewer that this point must be clarified along the manuscript. During the early section of the manuscript (Figure 1 and 2) we needed to ensure that the DNA repair defects observed in the absence of Cdc14 were not due to an indirect effect caused by the cell cycle role of the phosphatase in mitotic exit. The pre-synchronization with alpha factor before inducing the HO break ensures that all cells start the experiment at the same cell cycle stage, ruling out cell cycle interferences. A statement commenting on this subject has been incorporated in the text (line 94). Since in subsequent experiments (see for example figure 4A) we observed that the DNA repair defects in the absence of Cdc14 activity followed the same kinetics when inducing the HO break from asynchronous cultures, and due to the complexity of the alpha factor block and release approach when working with 24h experiments, we decided to perform these experiments from asynchronous cells. There is an exception in figure 7, where the use of pre-synchronized G1 cells allowed us to better follow the dynamics of Dna2 phosphorylation during the induction of the HO break. A small paragraph has been included in the text to justify the needed of G1 pre-synchronized cells under these particular experimental conditions (lines 314-315).

2) *“MINOR: What does it mean when the genome coverage data was normalised both against $t=0$ and 120 kb region on chromosome V? Please explain the logic/rationale here. And what does “normalised” mean in those context? Division? Subtraction?”*

In most cases, RPGC normalization during the alignment of the reads into the reference genome is enough for an accurate coverage measurement. However, we reasoned that the extensive resection observed in *cdc14-1* cells might affect the coverage of chromosome III when normalizing by RPGC only. We think that a second round of normalization using an intact chromosome V region corrects for this possible bias and, importantly, makes more reliable the comparison between different experiments and strains with a different degree of chromosome III resection. At all times normalization means division. We have included a paragraph in the method section describing the benefits of this normalization strategy (lines 831-838).

3) *“MINOR: Fig 1 BC. It would help readability greatly if the small horizontal cartoons/diagrams with the blue circles/red bars in them were labelled with text that describes what they are. e.g. Uncut parent, gene conversion (GC) products, etc.”*

We agree with the reviewer that due to the complexity of these Southern blots, it is difficult to interpret the GC outcomes obtained. To facilitate the interpretation of the blots we have now used a color code in figure 1A that facilitates the identification of the bands on previous Figure 1B (now figure 1C). Moreover, we have added a new diagram (Figure 1B) depicting all possible GC outcomes, their directionality, the distribution of the *Eco*RI and the band size expected for all of them. We have also incorporated the size of the GC products on the cartoons beside the Southern blots to facilitate their identification.

“Secondly, the juxtaposition/interleaving of the low and high exposure images—especially when the cropped areas are different makes it very challenging to understand the image. I think that the long exposure alone (without any cropping of the gel) would make the raw data a lot easier to understand.”

We included previously high and low exposure films due to the great differences in the intensity of the bands from the different GC outcomes. Still, we agreed that a single film facilitates the comparison between the bands and their distribution in the blot. We have now included a single film in figure 1C to illustrate the DNA repair deficiencies of *cdc14-1* cells.

4) *“MINOR: Fig 2. What explanation do the authors have for why the coverage for the most proximal probes returns to a higher level before that of the probes furthest away from the DSB? What repair mechanism may account for this?”*

It is precisely this observation what suggested us that the PMV strain was performing SDSA (Synthesis Dependent Strand Annealing) to repair the HO break. This observation was later supported by the GC analysis shown in figure 8. During SDSA, the formation of the D-loop ensures the copy of the genetic information from the donor sequence, that after its disengagement, re-anneals to the broken chromosome to prime DNA polymerization at both sides from the HO break. This polymerization is extended from the HO site outwards in both directions. Thus, proximal probes to the HO site will recover the signal earlier than those probes located distally from the break. We have included a sentence in figure legend 2B to comment about this observation (lines 597-599).

5) *“MINOR: Fig 3B. The presentation style here is unhelpful and distracting. Whilst a heatmap might be useful, the use of the faux/angular peaks/shading has no merit or utility. A simple heatmap of coverage through time would be much easier to visually understand.”*

We agreed with the reviewer that the angular shadows of these graphs impair the correct visualization of the data. We have now plotted the coverage data in a simple way by just using a flat color heatmap without shadows and triangles.

6) *“MINOR: Fig 3C. Line 533. Has it been demonstrated that this assay for coverage ± 100 nt flanking the HO cut site is indeed a measure of short-range resection? It seems to this reviewer that cleavage alone (without any resection) might reduce the probability of generating coverage in this region because the genome would be broken, and thus prevent any molecules/reads from spanning the junction. To understand this, has the assay been performed in conditions in which resection is completely inhibited (exo1D, dna2 mutant and sae2D?). If so, what were the results?”*

I completely agree with the reviewer about this subject. Just the cleavage of the HO site should be enough to reduce the coverage at the HO proximity. However, there are two observations that sustain the idea that short-range resection is indeed an active process. First, short-range resection at the cleavage site renders values of coverage close to zero. One would expect to have certain levels of coverage from those reads annealing just beside the HO break. Second, short-range resection is reduced at both sides of the HO break in the absence of Exo1, indicating that this nuclease actively participates in this process (Ramos F., et al., Cell Reports, 2022). Thus, although I agree that the cleavage of the HO site by itself might endorse a reduction in coverage levels at the HO vicinity, as the reviewer argues, it is also true that short-range resection seems to be a functional process taking place during the initial stages of the repair process.

“Note that, at most, resection here should presumably only reduce the coverage to 50% of maximum (based on loss of one strand), whereas graphs appear to show coverage dropping below 0.5 and close to zero in proximity of the DSB.”

We have sequenced the genome of cells following the Illumina protocol, so that after random fragmentation of genomic DNA, only double-stranded fragments are incorporated into the sequencing libraries. Thus, coverage at regions highly enriched in ssDNA due to resection (i.e. near the HO cleavage site) drop to levels close to 0, due to the lack of dsDNA to be cloned in the libraries from these particular areas. This explains the presence of coverage levels below 50%. To avoid confusion, we have included a statement in the manuscript clarifying this concept (line 151-153).

7) *“MINOR: Fig 3E. Line 542. How was the “maximum extent of resection” calculated/defined?”*

Maximum extent of resection was calculated by measuring the distance from the HO-induced break at which the read coverage reached a cut-off value of 0.9 relative to the whole chromosome coverage. We have included this information in the methods section (lines 839-840). We have also extended the information at figure legend 3E to clarify this subject (line 637-638).

8) *“MAJOR: Fig 6A Lines 240-246. Any differences in wild type \pm nocodazole or alpha factor were very hard for this reviewer to see.”*

We have now repeated the experiment three times and the averaged bands intensity have been plotted into a new graph (Figure 6B). Note that bands intensity for the distal probes (21 kb and 27 kb; used to check for over-resection) retrieve a 0.3 value when cells are blocked in metaphase (nocodazole), compared to a 0.6 value when cells are allowed to pass through mitosis (alpha factor). Although these differences might seem not substantial, we must consider that wild type cells in the presence of nocodazole still have a 30% of Dna2 foci dissolution (figure 6C), probably because of the effect of Cdc14 liberated during the DNA damage response (see next point). It is possible that the effect of DDR Cdc14 might account for reducing the differences between both conditions.

“Secondly, given that Cdc14 inhibition has the same effect in both arrests, I am unclear how the authors then conclude that Cdc14 mainly inhibits resection during its mitotic activation. This part needs substantial revision/clarification to make sense.”

I agree that this statement must be explained in a clearer way. The differences in bands intensity of probes distal from the break (21 kb and 27 kb; used principally to determine over-resection) between *cdc14-1* and wild-type in the presence of nocodazole are very subtle (note that *cdc14-1* is approximately 0.2, while in the wild type is about 0.3, both at 12h, now in figure 6B). Since in the presence of nocodazole only DDR Cdc14 is activated (see new figure 6A), this slight increase indicates that the liberation of Cdc14 during the damage response in wild-type cells has a very subtle effect in resection inhibition. However, the differences between *cdc14-1* and wild type in the presence of alpha factor are more marked (note that *cdc14-1* still approximately 0.2, while in the wild type is about 0.6, both at 12h, now in figure 6B). Since in the presence of alpha factor there is activation of mitotic Cdc14 (see new figure 6A), these results indicate that Cdc14 liberated by the FAR and MEN is more proficient than DDR Cdc14 at restraining resection. Still, we have been very cautious along the manuscript to make clear that although mitotic Cdc14 is the main responsible for resection inhibition, Cdc14 liberated during the damage response has also a slight contribution in this process (see for example lines 276-278).

To facilitate the understanding of these experiments, we have started the section by describing the differences between mitotic Cdc14 and DDR Cdc14 activation (lines 261-265). We have also clarified how the use of nocodazole or alpha factor can be used to determine the particular stage at which Cdc14 is inhibiting resection (line 268-280). Finally, we have incorporated a diagram in figure 6A to explain in a visual way how the addition of nocodazole or alpha factor affects the liberation of Cdc14 during the repair of the HO-induced break.

9) *“MINOR: Fig 6B. Lines 352. Please clarify how the lack of discordant pairs between II and VIII validates the assay? What is special about the III-VIII junction? Can a diagram of this locus/junction be provided to explain this control?”*

I agree that this sentence needs clarification. We just wanted to show that discordant read pairs were specifically accumulated between the two chromosomes involved in the repair of the HO break, the recipient (Chr. III) and the donor (Chr. V) (Please, see new diagram in Supp. Fig. 6E). If we repeat the same analysis but changing Chr. V for other chromosome not involved in the HO repair (i.e. Chr. VIII), then the Chr.III-Chr.VIII analysis does not retrieve discordant reads associated to these chromosomes, validating the specificity of the approach. We have now rewritten the sentence in a clearer way to avoid misleading (line 395-398).

10) *“MAJOR: Fig 8C is unintelligible. I am sure there are interesting data here, but the figure is too poorly explained to be understood by this reviewer. What purpose do all the inset cartoon chromosome bars serve/indicate?”*

We have improved figure 8C to clarify the data included in the graph. The *MATa* and *MATa'* profiles have been labelled in the figure to be properly identified. We have labelled the HO and HO-*inc* axis to facilitate the visualization of the distribution of the discordant read pairs. We have added a definition of “discordant reads pairs” (lines 385-387). We have included in the figure legend a description of the inset cartoons displaying the expected GC boundaries when assessing discordant read pairs (lines 749-752). We have also enriched the main text with a more detailed description for the data enclosed in figure 8C (line 403-406).

11) *“MAJOR: Line 364. Please explain how the directionality (and especially extent) of GC can be inferred from the coordinates. This sounds interesting/exciting, but the lack of description here prevents*

any critical understanding/appraisal. How long were the reads? Was there allele information across the entire fragment, or are their (expected) gaps on individual molecules between the (short) paired reads? Does this latter effect affect the data/analysis?"

The directionality and extent of GC can be inferred from the intersected *MATa-MATa'* peaks obtained when mapping the two reads from each discordant read pair. To facilitate the comprehension of this approach, we have implemented the schematic representation previously displayed in Supp. Fig. 6A (now in Supp. Fig. 6E) and extended the main text (lines 410-416).

Reads are 75 nt long and they do not cover the entire fragment cloned in the library when sequencing it. Because of that, we have to infer the position of the GC boundaries based on the intersection of the discordant read pairs peaks obtained during their alignment. This notion has been incorporated in the new diagram in Supp. Fig. 6E (as grey lines connecting blue and red discordant reads from each pair) and explained in the figure legend (lines 118-123 in supplementary figures).

We do not believe that the presence of unsequenced DNA gaps during paired end sequencing might affect the conclusions of the results presented herein. The generation of the coverage profile from the 75 nt discordant reads simply map the region where these sequences are enriched. In fact, the generation of longer reads could compromise the accuracy of the assay, since small reads have more probability to precisely map in the converged regions. Please, note that a particular read crossing the GC boundary will not align with the reference genome, since only those reads nearby the boundary, but entirely hybridizing with chromosome III or V (*MATa* or *MATa'*), are recovered from the bioinformatic approach.

12) *"MAJOR: Fig 8E. Lines 370-386. I am not convinced there are any differences \pm Cdc14 here. No statistics are provided. Moreover, even if they were, there appears to be no hypothesis being tested. The fact that some regions (may) show differences, and others not, may just be experimental noise. Without a hypothesis, multiple-test-corrected statistical thresholds need to be employed."*

We have now included a statistical test to validate the unbalanced distribution of GC events in the absence of Cdc14 activity. The 10% reduction in discordant read pairs detected distal to the left of the HO (Fig. 8E, A in the graph) in *cdc14-1* cells reaches statistical significance when compared to wild type cells, confirming that the phosphatase is required for extending GC tracts. We have also statistically tested for a net increase in B-E in the absence of Cdc14, corroborating the formation of shorter GC events (Fig. 8E, B-E in the graph). These data have been represented in a new graph, now in figure 8E. A sentence in lines 434-436 has been incorporated in the main text and in the figure legend. Please, note that a 10% difference, although it might seem a low value, is an important effect when dissecting GC distribution. Similar percentages than this presented herein for *cdc14-1* cells have been obtained when assessing essential factors involved in the dissolution of recombinant intermediates (i.e., Ira G., et al. Cell. 2003; Mitchel K. Plos Genetics. 2013).

Response to Reviewer #2:

Questions to be addressed

1) *“Cdk1 controls the localization of Dna2. In cells that lack Cdc14 phosphatase activity, Dna2 localization should remain in the nucleus throughout the cell cycle. A clear analysis of Dna2 localization without any DSB is needed for control by Cdc14.”*

We have now included figure 7G showing that wild-type cells accumulate Dna2 in the nucleoplasm in metaphase (nocodazole treatment) but not in G1 (alpha factor), indicating that the nuclease is subjected to subcellular changes during mitosis to G1 transition in an undamaged cell cycle (lines 328-331). We have also shown that in anaphase-blocked *cdc14-1* cells Dna2 remains nuclear (Figure 7G), suggesting that Cdc14 also targets Dna2 in non-DNA damage conditions during its mitotic activation (lines 331-333).

2) *“The new role of Cdc14 phosphatase is proposed in cells arrested in the G2/M phase in response to DNA damage. Cdc14 is known for its role in mitotic exit/anaphase. Is it known that Cdc14 is active earlier in G2/M arrested cells upon damage response?”*

It is already known that DDR-dependent liberation of Cdc14 in response to a DSB participates in specific stages of the repair process. We have emphasized this concept in the introduction (lines 57-61). We have also clarified that mitotic Cdc14 is the main responsible for promoting Dna2 dephosphorylation and resection inhibition (line 70).

3) *“Please provide more information on the *cdc14-1* allele; how do cells arrest or respond in general to increased temperature but without a DSB? Do they arrest and replicate normally?”*

I agree that this information must be added to the manuscript. It has been shown that in non-DNA damage conditions, *cdc14-1* cells arrest in late anaphase due to its incapacity to promote Cdk inactivation and mitotic exit. We have included a sentence containing this information in the manuscript (lines 261-263).

“In Figure S1, it would be good to indicate when the temperature is raised to 33°C.”

We have now indicated in the FACS profile the time point where cultures were risen to 33°C.

*“Would cells stay arrested in *cdc14-1* at high temperatures? How this affects the conclusions?”*

Since Cdc14 is required to promote Cdk inhibition, lack of the phosphatase induces a permanent arrest in late anaphase under non-DNA damage conditions. However, we have shown that *cdc14-1* cells in response to a DSB arrests in metaphase (Supp. Fig. 1C) with a permanent activation of the DNA damage response (Supp. Fig. 1B) due to the incapacity of these cells to timely repair the DSB (Figure 1). This means that the continued block in 2c observed in the mutant under our experimental conditions (Supp. Fig. 1A) is not due to the intrinsic role of the phosphatase in mitotic exit, but in DNA repair. Having said that, those *cdc14-1* cells that end up repairing the DSB and re-entering in the cell cycle will accumulate in anaphase due to the restrictive temperature. But even for those cells, we still measuring the accumulation of their repair products.

4) *“Line 122 and further. D-loop intermediate measurement – this assay measures initial DNA synthesis by PCR, as PCR product is detected only after initial DNA synthesis. It is interesting that initial DNA synthesis is completed, but further synthesis is not possible. This should be discussed. Also, the analysis should be done rather by qPCR to see the possible difference in *cdc14-1* cells at the nonpermissive temperature.”*

I agree with the reviewer that while DNA re-synthesis is impaired in the absence of Cdc14, D-loop extension does not seem to be compromised. These results indicate that Cdc14 might also be operating in the dissolution of the D-loop and/or the priming of DNA re-synthesis. We have commented about this hypothesis in the discussion section of the manuscript (lines 517-524). Besides, we must be cautious with the interpretation of this result, since the analysis of ssDNA at the HO boundary in *cdc14-1* cells retrieved values slightly below 0.5 (Fig. 3F), indicating that these zones are unstable at late time-points, probably due to the exhaustion of the RPA pool as a consequence of the extremely long resected tracts generated in the absence of the phosphatase. This effect might also affect subsequent stages of the repair process, as

the DNA re-synthesis step. This concept has also been commented in the results (lines 174-177) and in the discussion section (lines 529-530).

We have performed qPCR to determine the dynamics of D-loop extension in wild type and *cdc14-1* cells. The kinetics in the appearance of the amplification product is practically identical between both strains. We have included a graph in Supp. Fig. 1G containing the average data from two independent experiments (line 135 in the main text and lines 22-23 in supplementary figures).

5) *“The lack of repair is not due to excessive resection in *cdc13-1* as resection is comparable in WT and *cdc14-1* cells – Figure S1D. Here resection was measured in the absence of DSB repair, and it seems to be the same or even slower in *cdc14-1*. It would be good to plot WT and *cdc14-1* on one plot to see the resection slowness of *cdc14-1*.”*

As suggested by the reviewer we have plotted wild-type and *cdc14-1* cells on the same graphs to better visualize the similarity in the kinetics of resection between both strains. Please, note that in both wild-type and *cdc14-1* backgrounds there is a clear over-resection in the absence of repair (now in Supp. Fig. 1D, i.e. 21 kb band). This effect is expected in our model, since lack of DNA re-polymerization in a non-repairable HO break will avoid checkpoint deactivation and mitotic entry, thus preventing mitotic Cdc14 activation and resection inhibition.

*“Line 166 should say, “over-resect the two sides of repairable HO break. In nonrepairable break, there is no over-resection in *cdc14-1*.”*

We have corrected the sentence according to the reviewer’s suggestion (line 183).

*“Dna2 depletion in DNA2-AID *cdc14-1* stain rescues the repair defect. Would SGS1 deletion (*Sgs1* works with *Dna2* in resection) suppress *cdc14-1* in a similar way as DNA2-AID? It is an important experiment to validate the model.”*

I agree with the reviewer that deletion of *Sgs1* should bypass the over-resection phenotype on *cdc14-1* cells. However, we have previously observed that a *sgs1Δ* mutant generated in the PMV background retrieved also a marked over-resection phenotype (Ramos et al., Cell Reports, 2022). This observation is consistent with the resection profile generated in this strain (new Supp. Fig. 4D, left panel). We did not observe a clear rescue of the over-resection phenotype of *cdc14-1* cells when disrupting the helicase (Supp. Fig. 4D, right panel), probably due to the inherent hyper-resection of *sgs1Δ* cells in the PMV background. We have commented on this subject in the manuscript (lines 225-227).

6) *“The authors suggest that excessive resection is toxic in *Cdc14* minus cells, but it could be that more resection is observed because there is no repair, and less resection can bypass the toxicity of *Cdc14* depletion (whatever the reason is). To test this possibility, it would be good to limit resection in a different way, such as *fun30* deletion.”*

I completely agree with the reviewer. Although it looks clear that Cdc14 negatively controls resection progression by targeting Dna2, we cannot discard other functions for the phosphatase in the repair pathway. Moreover, since Dna2 degradation does not completely restore the over-resection observed in *cdc14-1* cells, it is possible that the phosphatase might be controlling other stages of the repair process. Considering the number of Cdk targets identified in the damage response during the last years, it is quite possible that Cdc14 could have multiple roles in DSB repair. We have included a full paragraph commenting on these ideas in the discussion section (lines 505-524).

Still, we believe that the targeting of Dna2 by Cdc14 is an active mechanism that controls resection length, since the deletion of Exo1 in *cdc14-1* cells did not restore the over-resection phenotype (Fig. 4B). It is important to note that the *exo1Δ* mutant is even more affected in resection progression than the *sgs1Δ* mutant in PMV background (see Ramos et al., Cell Reports, 2022). To validate even further this observation, and as suggested by the reviewer, we have now tested resection dynamics of *cdc14-1* cells in the absence of Fun30, and the data have been included in a new supplemental figure 3C. The elimination of Cdc14 activity also promote over-resection in cells lacking Fun30, indicating that Cdc14-dependent resection inhibition is specific of the Sgs1-Dna2 pathway (lines 204-206 in the main text and lines 61-64 in supplementary figures).

7) *“Line 208. Dna2 is not activated in response to Cdk-dependent phosphorylation. It is simply phosphorylated by Cdk1 in a cell cycle-dependent manner. This phosphorylation regulates its nuclear localization and in consequence, recruitment to DSBs.”*

We thank the reviewer for the clarification. We have added a sentence in the text correcting this issue (lines 232-233).

Minor

1) *“The topic of the manuscript is complex and needs attention to describe experiments in a simpler, easy-to-follow way.”*

I agree with the reviewer that the manuscript is complex and that some experiments and approaches were difficult to follow. By taking into consideration all suggestions and advice from the three reviewers we have significantly improved the readability of the manuscript. We have changed expressions, incorporated new sentences, and modified some paragraphs to make the article more accessible to non-expert readers. We have also improved the figures and incorporated new schematic representations that we believe, will facilitate the comprehension of the experiments presented in this work.

2) *“In the introduction line 40, please add information on the role of Dna2 phosphorylation in the regulation of nuclear localization (Kosugi et al., 2009).”*

We have added that Cdk-dependent phosphorylation of Dna2 induces the transport of the nuclease from the cytoplasm to the nucleus. We have also mentioned the work of Kosugi et al. (line 40)

3) *“Fig 1AB is hard to follow. Perhaps adding the sizes of all possible products in the diagram and indicating which one is which in B would be helpful.”*

We agreed with the reviewer that due to the complexity of these Southern blots, it is quite difficult to interpret the multiple bands obtained. To facilitate the interpretation of the blots we have now use a color code in Figure 1A to easily identify the bands on previous Figure 1B (now figure 1C). Moreover, we have also added a new diagram (Figure 1B) depicting all possible GC outcomes, their directionality, the distribution of the *Eco*RI, and the precise size of the band expected for all of them. Moreover, as proposed by reviewer#1 we have specified the size of the GC products on the cartoons beside the Southern blots and we have used a single film exposure to make the data easier to follow.

4) *“Line 101 – indicate “initial” resection, as the disappearance of the cut band only informs about initial resection.”*

As suggested by the reviewer we have indicated “initial” resection when referring to the disappearance of the “Cut” band (line 111).

5) *“Lane 110 – please modify the second part of the phrase. Do you mean “later stage of repair” or “completion of repair”? As stated, it is not clear.”*

We have followed reviewer’s recommendation and have changed this sentence, indicating that Cdc14 is not required for DNA end resection initiation but for subsequent steps of the recombinational repair pathway (line 119-121)

6) *“Resection during repair in a strain nearly identical to the one studied here was analyzed in a similar way by Chung et al. 2010 (PMID: 20485519). These results should be discussed.”*

We have added a sentence to remark that a similar strategy was used by Chung et al. to determine the kinetics of resection/repolymerization in a similar inter-chromosomal DNA repair system. We have mentioned that the kinetic of repair in this system was nearly identical to that observed in the PMV strain (line 137-138).

7) *“The rationale for an experiment in 6A should be better explained for people who do not study Cdc14.”*

I agree that due to the complexity of these experiments, a better explanation for the reasons behind this approach is required. First, we have started this section by explaining that Cdc14 is activated in mitosis (FEAR and MEN) and in response to DNA damage (lines 261-265). Second, we have clarified how the use of nocodazole or alpha factor can be used to determine the particular stage at which Cdc14 is inhibiting resection (line 269-270). Third, we have included a diagram (now in figure 6A) to explain in a visual way how the use of nocodazole or alpha factor give us information about the implication of DDR Cdc14 or mitotic Cdc14 in resection inhibition.

Response to Reviewer #3:

Major points

1) *“Reproducibility. For the majority of the experiments shown in the paper, it was not entirely clear if they were replicated. In fact, lack of statements on biological replication suggests they are not. While I can see that in a major study such as this not all experiments can be replicated, this needs to be done at least for key experiments. This is especially true, since not all effects are as “black-and-white” as the experimental description suggests.”*

We have now repeated all the experiments shown in the manuscript for at least two times. A statement in the figure legends specifying the number of repetitions for each particular experiment has been included.

2) *“The effect of the CDC14 mutant is intriguing, but they are only shown for a single system at a single genomic location. The text of the paper does however suggest that these findings can be generalized. At least the hyper-resection phenotype should be reproduced at another genomic location or alternatively the text would need to be strongly adapted to this shortcoming.”*

I agree with the reviewer that with the data enclosed in the previous manuscript we could not generalize the over-resection phenotype of *cdc14-1*. To sort this problem out, we have induced an HO break outside the *MAT* region at the *LEU2* locus on chromosome III in the YMV80 background (Vaze et al., 2002, Molecular Cell). In this background, the HO break is repaired by break-induced replication (BIR) or single strand annealing (SSA) recombinational DNA repair pathways. We have designed a new approach to analyze over-resection by Southern blot in this background (new figure 2C). Interestingly, we have retrieved a similar over-resection phenotype in cells lacking Cdc14 activity (new figure 2D). These results not only extend the role of Cdc14 in controlling resection length to other genomic locations but also to other recombinational DNA repair systems (line 142-146)

3) *“The resection assays in Fig. 2, 4, 5 and 6 are not easy to interpret for the non-expert as there is a biphasic response of resection and re-synthesis. As a strength of the system is that repair can be unambiguously quantified, I suggest to plot repair kinetics next to these graphs. Also, I miss plots of resection vs distance from DSB.”*

In these graphs, the repair kinetics are already represented under the term “0 kb”. However, I agree with the reviewer that this term does not properly illustrate the repair capacity of the HO break. To avoid confusion, we have changed this expression for “Uncut/Repair” (“U/R” in the figures) in all Southern blots, figures and legends along the entire manuscript. The resection profile for each distance from the DSB (0.7 kb, 3 kb, 6kb, 21 kb and 27 kb) is represented by the decay in the band signal intensity during the earlier time-points (0h-6h) from the HO induction.

4) *“The quantifications are often poorly labelled, particularly in Fig. 3. While I appreciate the use of cartoons as labelling, an additional labelling of the plots is needed. Additionally, figure legends need to be rewritten. They have sufficiently detail about experimental procedures, but they do not at all clarify what is being shown where. The combination of these two factors makes it often difficult to understand what is actually shown in which figure.”*

I acknowledge the reviewer for this observation. We have now re-labelled most of the plots to clarify what is being shown in the graphs. We have also started every figure legend with a small description to facilitate their identification in the main text/figures.

5) *“Figure 4 and 5 show a lot of pairwise comparisons in the individual figure panels and the authors do not clarify whether additional comparisons are possible. However, additional comparisons would be very useful, for example CDC14-1 vs CDC14-1 DNA2 add back (Fig. 5) or CDC14-1 DNA2-AID vs CDC14-1 DNA2-phosphomimetic (Fig. 5).”*

I agree with the reviewer that additional comparisons between strains is helpful for the correct visualization of the resection profiles. As suggested, we have added a *cdc14-1* vs *cdc14-1* Dna2-AID in supplemental figure 4C (line 223) and a *cdc14-1* Dna2-AID vs *cdc14-1* Dna2-AID pRS305-Dna2^{S17-237D} in supplemental figure 5 A (lines 255-257).

Minor points

6) *“Fig.8 I find the comparison of different time points problematic, T10 h for WT and T24 h CDC14-1. It is also not clear to me why those timepoints were chosen. According to Fig. 8A not much DNA synthesis happens in CDC14-1 after 10 h.”*

Due to slow dynamics in DNA repair observed in the absence of Cdc14, we decided to use the 24 h sample to attain the maximum levels of DNA repair. Still, I agree with the reviewer that there is little DNA re-synthesis from 9 h, suggesting that the levels of GC would be similar between 9 h and 24 h in the absence of the phosphatase. As expected, we have obtained a similar GC profile between both time-points in the *cdc14-1* background (new Supp. Fig. 6D). This result has been included in the text (lines 377-379).

7) *“Fig. 6A There does not seem to be much of a difference between the nocodazole and alpha-factor treatment of WT cells. This seems at odds with the authors conclusion.”*

To increase the reproducibility of this result, we have now repeated the experiment three times. The averaged bands intensities have been plotted into a new graph (Figure 6B). Note that the intensity for the distal probes bands (21 kb and 27 kb; used to check over-resection) retrieved a 0.3 value when cells are treated with nocodazole, compared to a 0.6 value when cells are treated with alpha factor. Although these differences might seem not substantial, we must consider that wild type cells in the presence of nocodazole still have a 30% of Dna2 foci dissolution (figure 6C), probably due to the effect of Cdc14 liberated during the DNA damage response. This effect might account for the rather small differences observed between both conditions. We have added a statement in the text commenting on this possibility (lines 276-278).

8) *“Fig. 4C There seems to be only a partial defect between EtOH and IAA treatment. Can the authors please comment on this?”*

I agree with the reviewer that there is a partial defect between EtOH and IAA treatment. We believe that this is due to the negative effect that the AID tag might exert over Dna2 activity. Accordingly, the comparison between an untagged *cdc14-1* strain and a *cdc14-1* Dna2-AID strain proposed by the reviewer (new Supp. Fig. 4C) clearly shows an enlarged difference between both strains. A statement about this possibility has been included in the text (line 224).

9) *“Fig. 4A-B I am not sure whether a comparison is possible, but if it is, it seems like the EXO1 mutant appears to have enhanced resection, which would be different from other findings.”*

I agree with the reviewer that the effect of Exo1 in resection initiation is hard to see in these blots, probably due to the prolonged time points taken between samples for this particular analysis. However, the role of Exo1 in the PMV strain has been well characterized previously in Ramos F., Cell Reports, 2022. In this paper the quantification of resection kinetics in cells lacking Exo1 was approximately half of that observed in the wild type. Still, to avoid confusion on this subject, we have now included a supplemental figure 3B, where we have compared the dynamics of resection between wild-type and *exo1Δ* cells in samples taken every 30 min. As previously reported, a *exo1Δ* strain displayed a clear reduction in resection proficiency and a stabilization of the resection intermediates, indicative of inefficient resection progression (line 201 in the main text and lines 55-60 in supplementary figures).

10) *“Fig. 1B-C In the graphs blue labelling of graph and legend appears not to match.”*

We have now used the same color for the labelling of the graph and legend.

REVIEWERS' COMMENTS

Reviewer #1 (Remarks to the Author):

REVISION 03-2023

Cdc14 phosphatase counteracts Cdk-dependent Dna2 phosphorylation to inhibit resection during recombinational DNA repair

The authors have answered most of my queries to an acceptable level. A few points/suggestions/comments remain that they may wish to take into account. Only one MAJOR (a methodological/clarification) point remains.

I congratulate them on their interesting methodological developments and the overall study.

(Point 3) MINOR: It would greatly simplify reading if the line graphs on the right-hand side of Fig 1C were ordered (from top to bottom) in the same order as the fragments on the gel.

i.e. 6.8, 2.9, 2.45, 2.3 kb

Currently the order is 6.8, 2.45, 2.3, 2.9 kb for no obvious (or helpful) reason.

The single blot is much easier to follow, however, I would perhaps have elected to use the stronger exposure so as to be able to observe all the bands being reported.

(Point 6) The reviewers did not completely answer my question, but the answer is sufficient. Nonetheless, I suggest it worth them performing (in the future) a sequencing experiment with DNA digested in vitro by HO, to determine the drop in coverage caused by 100% cleavage at HO (when there is no resection). Such an experiment may help with their future interpretations of this interesting methodology.

MAJOR: Nevertheless, I feel it is insufficient to refer to using the "Illumina protocol" to generate genome-wide data presented in this paper. Rather, in the methods, there should be a summary description of the general nature of this protocol. i.e. preparation of genomic DNA (how? By what protocol? How were cells sampled? How many? Were they fixed in any way prior to gDNA extraction?); sonication? (or was the DNA fragmented in another way?); Was random priming used? Or were dsDNA adapters ligated onto end-repaired fragmented DNA? These details are critical to the understanding and interpretation of the data, and for others to reproduce the observations and build upon them.

I had previously assumed the data used random priming...but I tend to infer that that is not the case (based on the author responses). However, the text is currently ambiguous and requires clarification.

Fig3 (MINOR); I only noticed upon revision of the improved Fig3B that "time" goes in a different direction in Fig 3B to Fig 3A, making comparison complex. (To make it even more cognitively challenging, time goes from left to right in panels C-F...)

The authors may wish to flip the panel order in fig 3A to be consistent with 3B (top to bottom).

(Point 7). MINOR: The explanation that maximum extent of resection refers to cut-off value of 0.9 relative to the whole chromosome coverage, doesn't seem to agree with the coloured heatmap data representation in Fig 3B. Again, this was another point that I could not adequately critique in the prior figure due to the prior presentation form.

I assume the cut-off means, coverage of 0.9 or less. Thus, this would include all regions in the heatmap that are not red (i.e. orange, yellow, green, cyan, blue, purple). For both strains, at all

but the earliest timepoint, this seems to go significantly further than reported in panels 3E. Based on the plots, perhaps they are using a cut-off of around 0.5? (i.e. somewhere in the middle of the green colour?)

(Points 9-12). I remain uncertain of the differences reported by the analyses performed in Fig 8. However, I think this is a provocative and interesting analytical technique, is not critical to the central message of the paper, and should be presented to the wider research community.

Reviewer #2 (Remarks to the Author):

The revised manuscript is improved and the authors addressed most of the questions of this this reviewer.

Reviewer #3 (Remarks to the Author):

The revised version of Campos et al. shows a role of the Cdc14 phosphatase in counteracting/restricting DNA end resection, acting at the level of phospho-regulation of the Dna2 nuclease.

The revised version of the manuscript addresses all points that I have previously raised and I find the manuscript strongly improved. I recommend it for publication.

Response to Reviewer #1:

1) *“(Point 3) MINOR: It would greatly simplify reading if the line graphs on the right-hand side of Fig 1C were ordered (from top to bottom) in the same order as the fragments on the gel. i.e. 6.8, 2.9, 2.45, 2.3 kb. Currently the order is 6.8, 2.45, 2.3, 2.9 kb for no obvious (or helpful) reason.”*

I agree with the reviewer’s suggestion. The new order suggested will facilitate the identification of the graphs within the figure. We have re-ordered the graphs accordingly.

“The single blot is much easier to follow, however, I would perhaps have elected to use the stronger exposure so as to be able to observe all the bands being reported.”

In the previous figure, we used a moderate exposition to avoid overexposure of the higher intense bands. Still, it is true that the low intensity bands were hard to see. Therefore, we have now followed reviewer’s recommendation and we have used a longer exposed film to be able to identify all the bands reported.

2) *“(Point 6) The reviewers did not completely answer my question, but the answer is sufficient. Nonetheless, I suggest it worth them performing (in the future) a sequencing experiment with DNA digested in vitro by HO, to determine the drop in coverage caused by 100% cleavage at HO (when there is no resection). Such an experiment may help with their future interpretations of this interesting methodology.”*

I completely agree with the reviewer that this experiment will be very valuable to determine the HO fingerprinting after cleavage. We will take his/her advise for subsequent analysis of short-range resection.

3) *“MAJOR: Nevertheless, I feel it is insufficient to refer to using the "Illumina protocol" to generate genome-wide data presented in this paper. Rather, in the methods, there should be a summary description of the general nature of this protocol. i.e. preparation of genomic DNA (how? By what protocol? How were cells sampled? How many? Were they fixed in any way prior to gDNA extraction?); sonication? (or was the DNA fragmented in another way?); Was random priming used? Or were dsDNA adapters ligated onto end-repaired fragmented DNA? These details are critical to the understanding and interpretation of the data, and for others to reproduce the observations and build upon them.”*

We have now improved the material and method section by incorporating all the suggestion raised by the reviewer. A description of the Southern blot sampling has been added to the genomic DNA extraction protocol (lines 554-555). We have also included a complete description of the protocol used for the generation of the genomic libraries (lines 591-598).

“I had previously assumed the data used random priming...but I tend to infer that that is not the case (based on the author responses). However, the text is currently ambiguous and requires clarification”

We have specified in the material and method section the precise procedure used to generate the genomic libraries. This protocol uses adaptors and index DNA sequences that facilitate the identification of the samples during the sequencing step. An explanation has been added to the material and method section of the manuscript (lines 591-598).

4) *“Fig3 (MINOR); I only noticed upon revision of the improved Fig3B that "time" goes in a different direction in Fig 3B to Fig 3A, making comparison complex. (To make it even more cognitively challenging, time goes from left to right in panels C-F...). The authors may wish to flip the panel order in fig 3A to be consistent with 3B (top to bottom).”*

I am completely agreed that the sampling order in figure 3A must be corrected to be consistent with the time points shown in figure 3B. We have re-ordered the time-points in figure 3A accordingly.

5) *“(Point 7). MINOR: The explanation that maximum extent of resection refers to cut-off value of 0.9 relative to the whole chromosome coverage, doesn't seem to agree with the coloured heatmap data representation in Fig 3B. Again, this was another point that I could not adequately critique in the prior figure due to the prior presentation form. I assume the cut-off means, coverage of 0.9 or less. Thus, this would include all regions in the heatmap that are not red (i.e. orange, yellow, green, cyan, blue, purple). For both strains, at all but the earliest timepoint, this seems to go significantly further than reported in*

panels 3E. Based on the plots, perhaps they are using a cut-off of around 0.5? (i.e. somewhere in the middle of the green colour?)”

I agree with the reviewer that this point needs clarification. The apparent disconnection between the total length of resection (Fig. 3E) and the colored heatmap data (Fig. 3B) comes from the fact that the average coverage of chromosome III diminishes with resection. Since the cut-off value for each sample is relativized against the chromosome III coverage average, this value is subjected to a decrease in those samples that accumulates ssDNA. This normalization corrects for the extensive lack of chromosome III coverage observed in the absence of Cdc14 activity. We have included a statement in the material and method section explaining this concept (lines 628-631).

6) *“(Points 9-12). I remain uncertain of the differences reported by the analyses performed in Fig 8. However, I think this is a provocative and interesting analytical technique, is not critical to the central message of the paper, and should be presented to the wider research community.”*

I am agreed with the reviewer that, although the differences in gene conversion between wild type and *cdc14-1* cells are not substantial, the presentation of this new approach specifically designed to identify GC boundaries constitutes on its own a great leap in the analysis of recombinational DNA repair that will be of great interest for the DNA repair community. I appreciate his/her comments on the design of our new methodological approach.